# Interpretable machine learning for high-dimensional trajectories of aging health

**Spencer Farrell**[1]*, **Arnold Mitnitski**[2,3†], **Kenneth Rockwood**[2,3], **Andrew D. Rutenberg**[1]*

**1** Department of Physics and Atmospheric Science, Dalhousie University, Halifax, Nova Scotia, Canada, **2** Division of Geriatric Medicine, Dalhousie University, Halifax, Nova Scotia, Canada, **3** Department of Medicine, Dalhousie University, Halifax, Nova Scotia, Canada

† Deceased.
* spencer.farrell@dal.ca (SF); adr@dal.ca (ADR)

**Data Availability Statement:** Our code is available at https://github.com/Spencerfar/djin-aging. The English Longitudinal Study of Aging waves 0-8, 1998-2017 with identifier UKDA-SN-5050-17 is available at https://www.elsa-project.ac.uk/

## Abstract

We have built a computational model for individual aging trajectories of health and survival, which contains physical, functional, and biological variables, and is conditioned on demographic, lifestyle, and medical background information. We combine techniques of modern machine learning with an interpretable interaction network, where health variables are coupled by explicit pair-wise interactions within a stochastic dynamical system. Our dynamic joint interpretable network (DJIN) model is scalable to large longitudinal data sets, is predictive of individual high-dimensional health trajectories and survival from baseline health states, and infers an interpretable network of directed interactions between the health variables. The network identifies plausible physiological connections between health variables as well as clusters of strongly connected health variables. We use English Longitudinal Study of Aging (ELSA) data to train our model and show that it performs better than multiple dedicated linear models for health outcomes and survival. We compare our model with flexible lower-dimensional latent-space models to explore the dimensionality required to accurately model aging health outcomes. Our DJIN model can be used to generate synthetic individuals that age realistically, to impute missing data, and to simulate future aging outcomes given arbitrary initial health states.

## Author summary

Aging is the process of age-dependent functional decline of biological organisms. This process is high-dimensional, involving changes in all aspects of organism functioning. The progression of aging is often simplified with low-dimensional summary measures to describe the overall health state. While these summary measures of aging can be used predict mortality and are correlated with adverse health outcomes, we demonstrate that the prediction of individual aging health outcomes cannot be done accurately with these low-dimensional measures, and requires a high-dimensional model. This work presents a machine learning approach to model high-dimensional aging health trajectories and mortality. This approach is made interpretable by inferring a network of pairwise interactions

accessing-elsa-data. This requires registering with the UK Data Service.

**Funding:** ADR thanks the Natural Sciences and Engineering Research Council (NSERC) for an operating Grant (RGPIN 2019-05888). KR has operational funding from the Canadian Institutes of Health Research (PJT- 156114) and personal support from the Dalhousie Medical Research Foundation as the Kathryn Allen Weldon Professor of Alzheimer Research. The funders had no role in study design, data collection and analysis, decision to publish, or preparation of the manuscript.

**Competing interests:** Author Arnold Mitnitski was unable to confirm their authorship contributions. On their behalf, the corresponding author has reported their contributions to the best of their knowledge. The authors have declared that no other competing interests exist.

between the health variables, describing the interactions used by the model to make predictions and suggesting plausible biological mechanisms.

## Introduction

Aging is a high-dimensional process due to the enormous number of aspects of healthy functioning that can change with age across a multitude of physical scales [1, 2]. This complexity is compounded by the heterogeneity and stochasticity of individual aging outcomes [3, 4]. Strategies to simplify the complexity of aging include identifying key biomarkers that quantitatively assess the aging process [5, 6] or integrating many variables into simple and interpretable one-dimensional summary measures of the progression of aging, as with "Biological Age" [7–9], clinical measures such as frailty [10, 11], or recent machine learning models of aging [12, 13]. Nevertheless, one-dimensional measures only summarize the progression of aging, and so can miss significant aspects of high-dimensional aging trajectories and of heterogeneous aging outcomes. We introduce a machine learning approach to model high-dimensional trajectories directly, while still learning interpretable aspects of our model through an explicit network of interactions between variables.

The increasing availability of large longitudinal aging studies is beginning to provide the rich data-sets necessary for the development of flexible machine learning models of aging [14]. Methods for predictive modelling of individual health trajectories of disease progression have already been developed [15–20], but they generally are not joint models that include both mortality and the progression of aging [20]. There has also been progress on learning interpretable summaries of aging progression [12, 13], generalizing biological-age approaches but still producing low-dimensional summaries of aging.

Less progress has been made on the more general problem of modeling high-dimensional aging trajectories. Stochastic-process joint models that simultaneously model longitudinal and survival data have been proposed [21–23], but have only been implemented for one or two health variables at a time. Farrell *et al.* [24] used cross-sectional data to build a network model that generated trajectories of 10 health variables and predicted survival, but it was limited to binary health measures.

In this work we use the English Longitudinal Study of Aging (ELSA, [25]), which is a large observational population study including a wide variety of variables with follow-up measurements for up to 20 years including mortality. Like other large observational studies, for most individuals it has many missing measurements, few irregularly-timed follow-ups, and censored mortality. Any practical approach to model such data must confront the challenges provided by missing and irregularly timed data and by mortality censoring.

While machine learning (ML) approaches can help us navigate these challenges with available data, they face additional challenges of interpretability [14, 26]. "Scientific Machine Learning" [27] or "Theory guided data science" [28] suggests that domain knowledge be used to constrain and add interpretability to ML models. For example, we require that aging is modelled as a network of interacting health components [29, 30], and that stochastic differential equations (SDEs) model the dynamical evolution of high-dimensional health states [21]. On the other hand we use general ML approaches to model survival or to impute missing data for baseline (initial) health states, where we may not be interested in interpretation.

The result (see Fig 1) is a powerful and flexible, but interpretable, approach to modelling aging and mortality from high-dimensional longitudinal data—one that preserves but is not crippled by the complexity of aging. We evaluate the resulting model with test data and

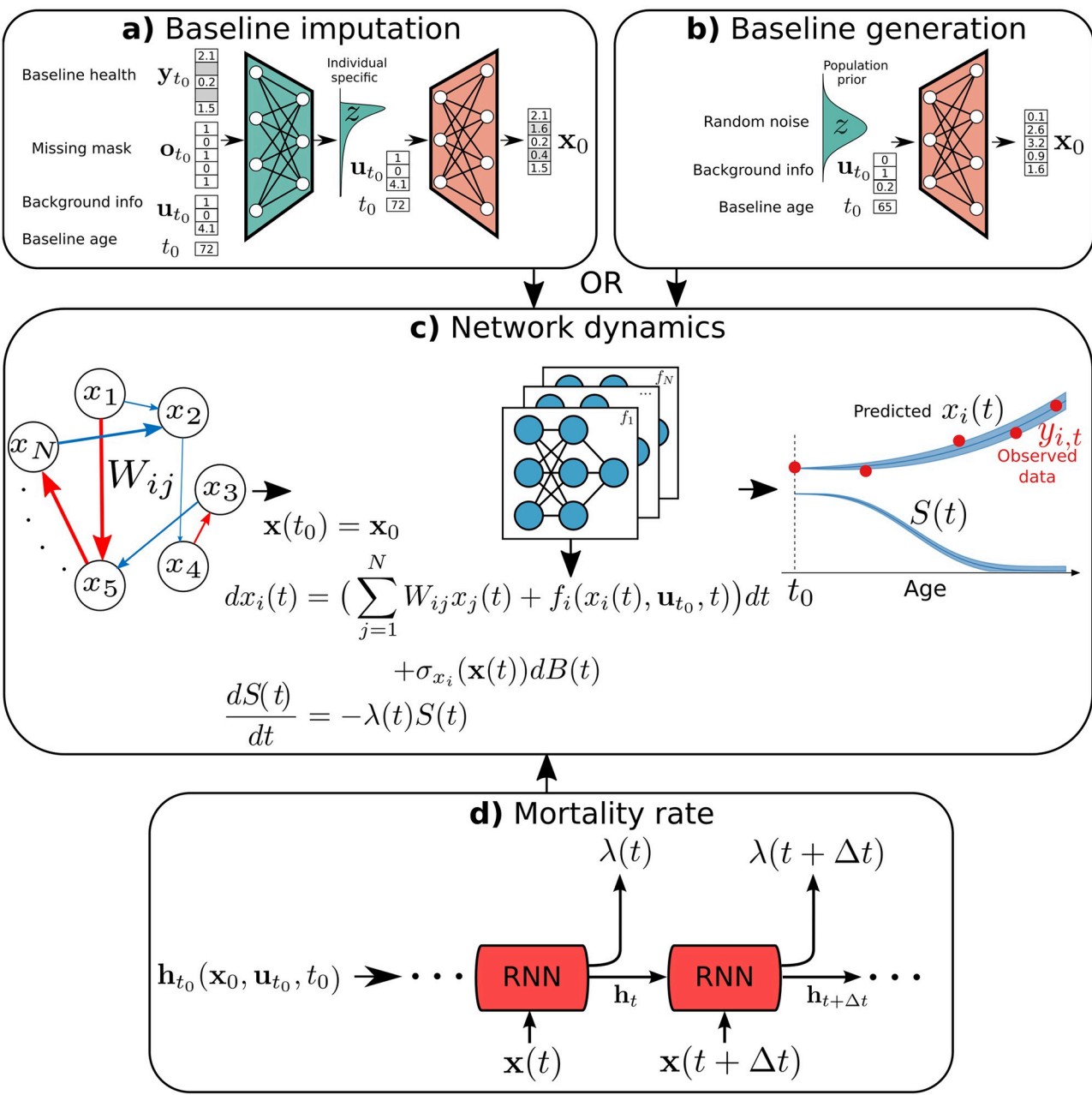

**Fig 1. DJIN model of aging. a)** Baseline imputation is performed using the baseline health measurement $\mathbf{y}_{t_0}$, missing mask $\mathbf{o}_{t_0}$, background health information $\mathbf{u}_{t_0}$, and baseline age $t_0$ as input to an encoder neural network (green) that parameterizes a latent distribution. Sampling from this latent distribution and using a decoder neural network (orange) gives an imputed complete baseline health-state $\mathbf{x}_0$. **b)** Baseline generation conditional on background health information $\mathbf{u}_{t_0}$, and baseline age $t_0$ can be used instead of imputation. The population latent distribution is sampled and used with the same decoder neural network (orange) to produce a synthetic baseline health state $\mathbf{x}_0$. **c)** Network dynamics stochastically evolve the health state $\mathbf{x}(t)$ in time starting from the baseline state $\mathbf{x}_0$. The stochastic dynamics are modeled with a stochastic differential equation which includes the pairwise network interactions with connection weight matrix $\mathbf{W}$, general diagonal terms $f_i(x_i(t), \mathbf{u}_{t_0}, t)$ parameterized as neural networks, and a diagonal covariance matrix for the noise $\boldsymbol{\sigma}_x(\mathbf{x})$ also parameterized with a neural network. **d)** The survival function evolves in time based on the state and history of the health state $\mathbf{x}$ using a recurrent neural network (RNN). The initial state of the RNN, $\mathbf{h}_{t_0}$, is set using the background health information $\mathbf{u}_{t_0}$, baseline age $t_0$, and $\mathbf{x}_0$. Details are provided in the Methods. The code for our model is available at https://github.com/Spencerfar/djin-aging.

compare with simpler linear modelling approaches. We use a variational Bayesian approach to infer the approximate posterior distribution of the both interaction network and individual health trajectories to approximate confidence bounds. We demonstrate our model's ability to robustly predict health trajectories using an interpretable network of constant linear interactions between health variables. Additionally, we demonstrate that flexible but low-dimensional latent space models of a similar structure cannot predict aging health outcomes as well as our high-dimensional DJIN model—confirming the high-dimensional nature of our approach and of the aging trajectories.

## Results

### ELSA dataset

We combine waves 0 to 8 in the English Longitudinal Study of Aging (ELSA, [25]) to build a dataset of $M = 25290$ individuals, with longitudinal follow-up of up to 20 years. In ELSA, self-reported health information is obtained approximately every 2 years and nurse-evaluated health with physical assessment and blood tests approximately every 4 years. Considering all waves together with 2 year increments, 27% of values are missing for self-reported variables, 78% of values are missing for nurse-evaluated variables, and 96% of individual mortality is censored. Training and test trajectories (see below) are sampled starting with baseline times starting at each of the waves; though at least one followup wave is required for test trajectories.

For a given starting wave, an individual's health state is observed at $K + 1$ times $\{t_k\}_{k=0}^{K}$ with a set of health variables $\{\mathbf{y}_{t_k}\}_{k=0}^{K}$. The vectors $\mathbf{y}_{t_k}$ describe the $N$-dimensional health state of an individual, where each of the $N$ dimensions represents a separate health measurement. We select $N = 29$ continuous-valued or discrete ordinal variables that were measured for at least two of the waves. Individuals also have background (demographic, diagnostic, or lifestyle) information observed at baseline, which is described by a $B$-dimensional vector $\mathbf{u}_{t_0}$. In principle, any baseline data can be used as background information. We select $B = 19$ continuous or discrete valued background variables. These are used as auxiliary variables at baseline; they aide the subsequent prediction of the health variables $\mathbf{y}_t$ vs time.

Variables used from the data-set were selected only by availability, not by predictive quality. All chosen variables and the number of observed individuals for each is shown in S1 Fig, the details of the variables are given in Table A in S1 Text.

### DJIN model of aging

We build a model to forecast an individual's future health $\{\mathbf{y}_{t_k}\}_{k>0}$ and survival probability $\{S(t_k)\}_{k>0}$ given their baseline age $t_0$, baseline health $\mathbf{y}_{t_0}$ and background health variables $\mathbf{u}_{t_0}$. It is a dynamic, joint, interpretable network (DJIN) model of aging. A schematic of our model is shown in Fig 1, while mathematical details are provided in the Methods.

Effective imputation is essential because none of the 25290 individuals in the data-set have a fully observed baseline health state. Fig 1a illustrates our method of imputation for the baseline health state. Variational auto-encoders have shown promising results for imputation [31, 32]. We impute with a normalizing-flow variational auto-encoder [33], where a neural network (green trapezoid) encodes the known information about the individual into an individual-specific latent distribution, and a second neural network (orange trapezoid) is used to decode states sampled from the latent distribution into imputed values. This is a multiple imputation process that outputs samples from a distribution of imputed values rather than a single estimate.

We have chosen this imputation approach because we can also use it to generate totally synthetic baseline health states given background/demographic health information and baseline age. Fig 1b illustrates this method. We randomly sample the prior population distribution of the same latent space used in imputation, and then combine this with arbitrary background information and use the same decoder as in imputation to transform the latent state into a synthetic baseline health state. With repeated random samples of the latent space, we generate a distribution of synthetic baseline health states.

Fig 1c illustrates the temporal dynamics of the health state in the model. Dynamics start with the imputed or synthetic baseline state $\mathbf{x}_0$. The health state is then evolved in time with a set of stochastic differential equations, similar to the Stochastic Process Model of Yashin *et al.* [21, 22, 34, 35]. The stochastic dynamics capture the inherent stochasticity of the aging process. We assume constant linear interactions between the variables, with an interpretable interaction network $\mathbf{W}$. This interaction network describes the direction and strength of interactions between pairs of health variables.

Fig 1d illustrates the mortality component of the model. The temporal dynamics of the health state is input into a recurrent neural network (RNN) to estimate the individual hazard rate for mortality, which is used to compute an individual survival function. Recent work shows that this approach can work well in joint models [20]. The RNN architecture uses the history of previous health states in mortality, otherwise mortality could only depend on the current health state and could not capture the effects of a history of poor health. We have chosen this RNN approach to mortality because it performs better than a feed-forward model with no history (as shown in S2 Fig).

We use a Bayesian approach to model uncertainty by estimating the posterior distribution of parameters, of health trajectories and of survival curves—as illustrated by the shaded blue confidence intervals in Fig 1C. To handle our large and high-dimensional datasets, we use a variational approximation to the posterior [36] instead of impractically slower MCMC methods. The variational approximation reduces the sampling problem to an optimization problem, which we can efficiently approach using stochastic gradient descent. Mathematical details are provided in the Methods. The code for our model is available at https://github.com/Spencerfar/djin-aging.

## Validation of model survival trajectories

We evaluate our model with test individuals withheld from training. Given baseline age $t_0$, baseline health variables $\mathbf{y}_{t_0}$, and background information $\mathbf{u}_{t_0}$ for each of these test individuals, we impute missing baseline variables and predict future health trajectories and mortality with the model. These predictions are compared with their observed values.

The C-index measures the model's ability to discriminate between individuals at high or low risk of death. We use a time-dependent C-index [37], which is the proportion of distinct pairs of individuals where the model correctly predicts that individuals who died earlier had a lower survival probability. Higher scores are better; random predictions give 0.5. In Fig 2a we see that our model (red circles) performs substantially better than a standard Cox proportional hazards model (green squares) with elastic net regularization and random forest MICE imputation [38, 39]. The horizontal lines show the C-index scores for the entire test set, and the points show predictions stratified by baseline age. Stratification allows us to remove age-effects in the predictions; we determine how well the model uses health variables to discriminate between pairs of individuals at the same age. Our model predictions do not substantially degrade when controlling for age, indicating that it is learning directly from health variables, rather than from age. Predictions degrade at older baseline ages due to the limited sample size.

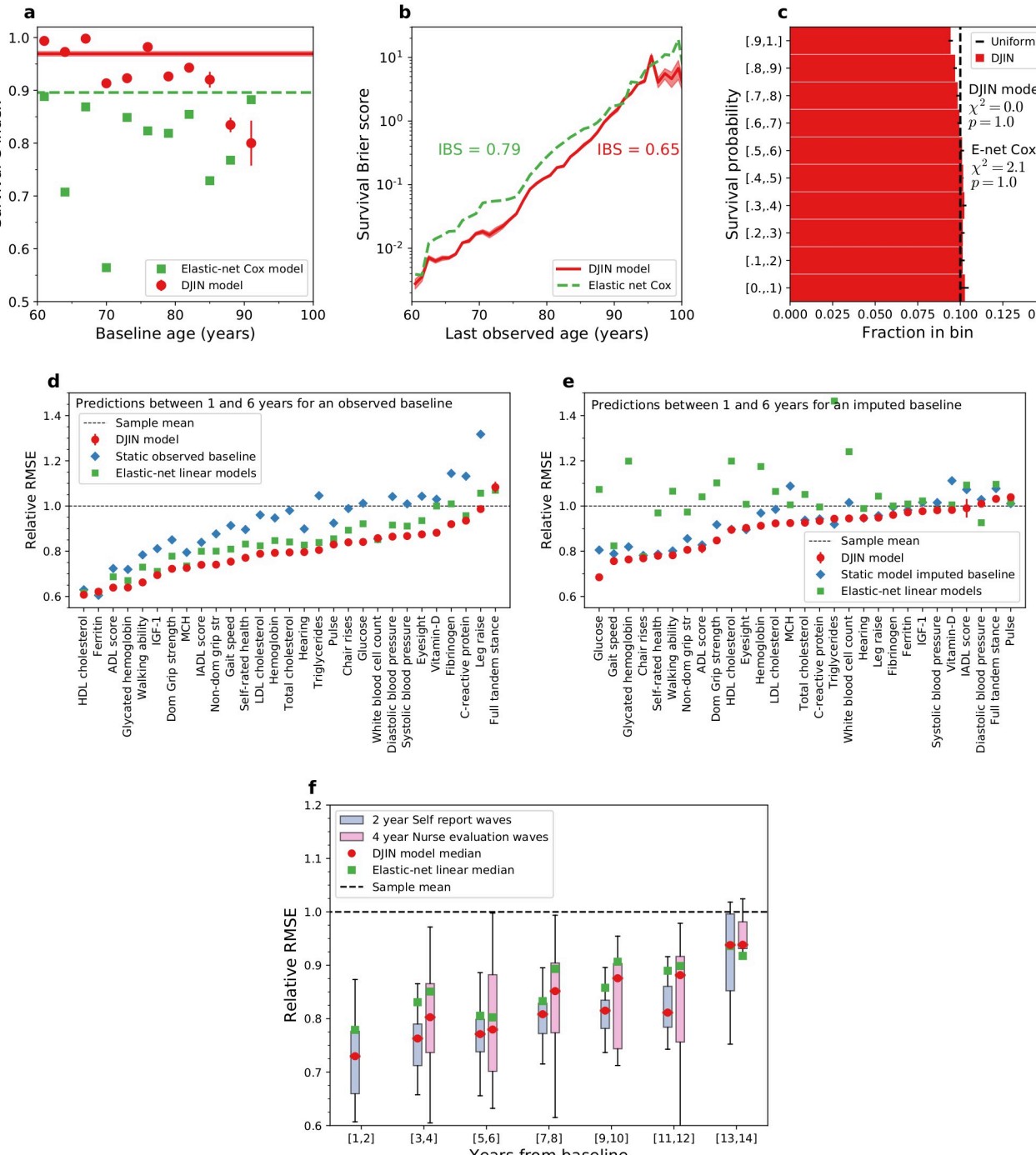

**Fig 2. Model predictions and validation.** Errorbars for all plots represent standard errors of the mean for 5 fits of the DJIN model. **a)** Time-dependent C-index stratified vs age (points) and for all ages (line). Results are shown for our model (red) and a Elastic net Cox model (green). (Higher scores are better). **b)** Brier scores for the survival function vs death age. Integrated Brier scores (IBS) over the full range of death ages are also indicated. The Breslow estimator for the baseline hazard is used for the Cox model. (Lower scores are better). **c)** D-calibration of survival predictions. Estimated survival probabilities are expected to be uniformly distributed (dashed black line). We use Pearson's $\chi^2$ test to assess the distribution of survival probabilities for our network model ($\chi^2 = 1.3$, $p = 1.0$) and an elastic net Cox model ($\chi^2 = 2.1$, $p = 1.0$). (Higher p-values and smaller $\chi^2$ statistics are better). **d)** RMSE scores when the baseline value is observed for each health variable for predictions between 1 and 6 years from baseline, scaled by the RMSE score from the age and sex-dependent sample mean (relative RMSE scores). We show the predictions from our model starting the baseline value (red circles), predictions assuming a static baseline value (blue squares), and 29 distinct elastic-net linear models trained separately for each of the variables (green squares). The DJIN predictions here come from the same model as for mortality and the elastic net Cox model is also a distinct model. (Lower RMSE is better). **e)** Relative RMSE scores when the baseline value for each health variable is imputed for predictions between 1 and 6 years from baseline. We

show the predictions from our model starting from the imputed baseline value (red circles), predictions assuming a static imputed value (blue squares), and predictions assuming an elastic-net linear model (green squares). (Lower RMSE is better). **f)** RMSE score distributions over all health variables for increasing years of prediction from baseline. The median RMSE score is shown as a black dotted line between the boxes showing upper and lower quartiles. Whiskers show 1.5x the interquartile range from the box. (Lower RMSE is better). Self-report and nurse-evaluated waves have distinct patterns of missing variables; nurse-evaluated waves have higher missingness overall.

We evaluate the detailed accuracy of survival curve predictions with the Brier score [40]. Individual Brier scores calculate squared error between the full predicted survival distribution $S(t)$ and the measured survival "distribution" for that individual, which is a step-function equal to 1 while the individual is alive and 0 when they are dead. Lower Brier scores are better, though the intrinsic variability of mortality will provide some non-zero lower bound to the Brier scores. In Fig 2b we show the average Brier score for different death ages for our model (blue) and a Cox model with a Breslow baseline hazard (green), indicating our model has a substantially lower error between the predicted and exact survival distributions for older ages (note the log-scale). The Integrated Brier Score (IBS) is computed by integrating these curves over the range of observed death ages, and highlights the improvement of predictive accuracy of our model as compared to Cox.

We evaluate the calibration of survival predictions with the D-calibration score [41]. For a well-calibrated survival curve prediction, half of the test individuals should die before their predicted median death age and half should live longer. Calibrated survival probabilities can be interpreted as estimates of absolute risk rather than just relative risk. The D-calibration score generalizes this to more quantiles of the survival curve, where the proportion observed in each predicted quantile should be uniformly distributed. In Fig 2c, we show deciles of the survival probability for our model (red bins), compared with the expected uniform black straight line. We compute $\chi^2$ statistics and p-values for the predictions of our model vs the uniform ideal, as well as for a Cox proportional hazards model (histogram in S3 Fig). Our model is consistent with a uniform distribution under this test ($p = 1.0$, $\chi^2 = 1.3$) as desired for calibrated probabilities. The Cox model is also calibrated ($p = 1.0$, $\chi^2 = 2.1$), but with a slightly worse $\chi^2$ statistic.

These results demonstrate that our DJIN model accurately predicts the relative risk of mortality of individuals (assessed by the C-index), predicts accurate survival probabilities (assessed by the Brier score), and properly calibrates these survival probabilities so that they can be directly interpreted as an absolute risk of death.

## Validation of model health trajectories

Model predictions of individual health trajectories are also evaluated on the test set. We compute the Root-Mean-Square Error (RMSE) for each health variable, and create a relative RMSE score by dividing by the RMSE obtained when using the age and sex matched training-set sample mean as the prediction. In Fig 2d, we show that the model (red circles) performs better than the age and sex-dependent sample mean (black dashed line) when the baseline value of the particular variable is observed. The RMSE here is computed for all predictions between 1 and 6 years from baseline. In Fig 2e we show that the model is predictive of future health values even when the initial value of the particular variable is imputed.

As measured by the relative RMSE, our model is better than a null model (blue squares) that carries forward the observed baseline (**d**) or imputed baseline value (**e**) for all ages. For comparison purposes, we also trained linear models with elastic net regularization and random-forest MICE imputation [38, 39] that have been trained separately to predict each health variable. We are therefore comparing our single DJIN model that predicts all 29 variables, to

29 independently-trained linear models. While the linear models perform better than the null model for observed baselines, our model performs better than both. For imputed baselines, the linear models with random-forest MICE imputation performs poorly even compared to the imputed null model, while our model continues to outperform both. In S4 Fig we show that our model only performs poorly when variables have a large proportion ($\gtrsim$ 90%) of missing values—though still better than linear models.

In Fig 2f, we show boxplots of RMSE scores over the health variables for 1–14 years past baseline, when the variable was initially observed at baseline. The model is predictive for long term predictions, and remains better than linear elastic net predictions for at least 14 years past baseline for the self-report waves (blue) and 12 years past baseline for the nurse-evaluated waves including blood biomarkers (pink).

In S5 Fig we show example DJIN trajectories for 3 individuals in the test set for the 6 best predicted health variables. We show both the mean predicted model trajectory and a visualization of the uncertainty in the trajectory. For comparison, the sample mean and elastic net linear model are shown. The predicted trajectories visually agree well with the data, and is often substantially better than either the elastic net linear predictions or the sample means for the corresponding variables.

These results demonstrate that our DJIN model predicts the values of future health variables from baseline better than standard linear models, and also better than population-mean or constant baseline models.

## Comparison with latent space models

In Fig 3 we compare the DJIN model, with dynamics directly in the high-dimensional space of observed health variables, with latent space models that have more flexible but less interpretable dynamics within a latent space of adjustable dimensionality. As illustrated by Fig 1, with details in Methods, these latent space models use dynamics directly on the initial latent states output by the VAE encoder $\mathbf{z}$. The dynamics of the latent variables use a full feed-forward neural network for the drift of the SDE. The latent trajectories $\mathbf{z}(t)$ are then decoded into predicted observed health states $\mathbf{x}_t$ with the VAE decoder. Since we can reduce the dimensionality of the latent space as compared to the space of observed health variables, we can investigate the effects of dimensionality. Since the latent-space model dynamics are not restricted to have only constant linear interactions, we can also investigate any limitations of the interpretable interactions imposed in the DJIN model.

Fig 3a shows that a large number of dimensions are required to accurately predict health trajectories. A one-dimensional model can be used to predict the relative risk of survival, as assessed by the C-index in Fig 3b. However, good predictions of the survival probability require at least two-dimensional models, as shown in Fig 3c. This suggests that single summary measures of aging such as biological age can capture the relative progression of aging, but cannot individually predict the specific heterogeneous health outcomes during aging [9, 13, 42].

In S6 and S7 Figs, we show in greater detail the results of 30-dimensional model and one-dimensional latent space models. Our DJIN model that only includes pair-wise linear interactions performs similarly to high-dimensional latent space models that use non-linear interactions. This suggests that our interpretable linear network approximation is sufficient for describing the dynamics of these variables. Linear pair-wise network approximation may work so well because we are interested in long term predictions, rather than short-time scale dynamics where the variables may be more strongly non-linearly coupled. Predictions with non-linear models may prove better with larger datasets, or with continuously acquired data that necessitate shorter timescales.

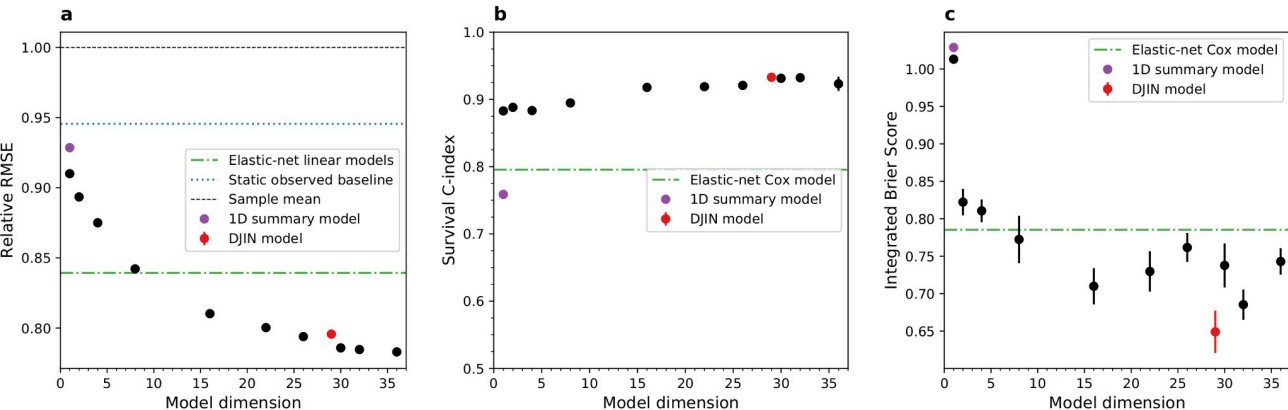

**Fig 3. Latent space model performance vs dimension.** Black points show latent space models of various dimension, red points show the DJIN model, green lines show the elastic net linear models. The purple point shows the 1D summary model described in S1 Text, which includes the information from the auxiliary background variables $\mathbf{u}_{t_0}$ within the latent state, rather than as a separate input in the model (see S7 Fig for more detail). Black points include $\mathbf{u}_{t_0}$ as a separate input in addition to $\mathbf{z}$. Points indicate the mean of 10 independent fits of the models, and error bars represent standard error of mean (often smaller than point size). **a)** RMSE for health predictions, relative to predictions with the population average (black dashed line). (Lower is better). **b)** Survival C-index. (Higher is better). **c)** Integrated Brier score for survival. (Lower is better).

## Validation of generated synthetic populations

Given baseline age $t_0$, and background information $\mathbf{u}_{t_0}$ for test individuals, we generate synthetic baseline health states and simulate a corresponding synthetic aging population. We evaluate these aging trajectories by comparing with the observed test population. We train a logistic regression classifier to evaluate if the synthetic and observed populations can be distinguished [18, 19, 43, 44]. We find that this classifier has below a 57% accuracy for the first 14 years past baseline (S8 Fig) – only slightly better than random. S8 Fig also shows that the DJIN model does better or equivalent to the 30-dimensional latent space model from Fig 3.

In S9 and S10 Figs we show the population and synthetic baseline distributions and population summary statistics for the trajectories vs age for ages 65 to 90. We find that our model captures the mean of the population, but slightly underestimates the standard deviation of the population (as expected due to our variational approximation of the posterior [36]). In S11 Fig we show the population synthetic survival function agrees with the observed population survival below age 90, where the majority of data lies.

The agreement of the synthetic and test populations demonstrates the DJIN model's ability to generate a synthetic population of aging individuals that resemble the observed population, though with slightly less variation.

We have made a synthetic population available at https://zenodo.org/record/4733386.

## DJIN infers interpretable sparse interaction networks

Our Bayesian approach infers the approximate posterior distribution of the interaction network weights; Fig 4 visualizes the network with the mean posterior weights. Weights with a 99% posterior credible interval including zero have been pruned (white)—all visible weights have posterior credible intervals either fully above or fully below zero. This cutoff is demonstrated in S12 Fig.

Connections are read as starting at the variable on the horizontal axis ($j$), and ending at the variable on the vertical axis ($i$), representing the connection weight matrix $W_{i \leftarrow j}$. Positive connections indicate that an increasing variable $j$ influences an increase in variable $i$. Negative connections mean an increasing variable $j$ influences a decrease in variable $i$. The interaction

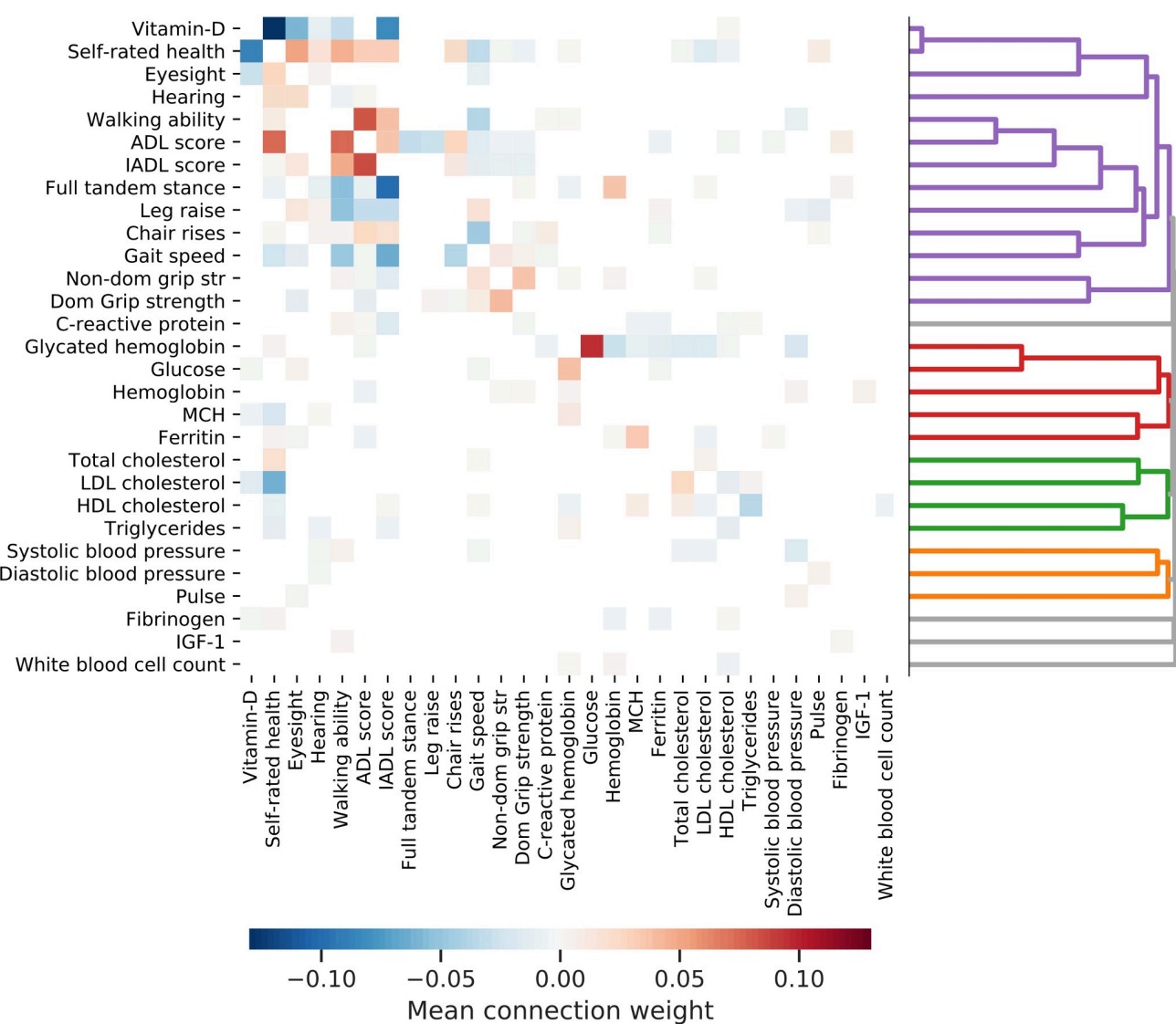

**Fig 4. Inferred interaction network.** Heatmap of the posterior mean value of the robust network weights. Weight directions are read from the horizontal axis ($j$) towards the vertical ($i$), $W_{i \leftarrow j}$. The sign and color of the weight signify the direction of effect—a positive weight implies that an increase in a variable along the horizontal axis influences the vertical axis variable to increase. A negative weight implies that an increase in a variable along the horizontal axis influences the vertical axis variable to decrease. Hierarchical clustering is applied to the absolute posterior mean of the robust weights to create a dendrogram (at right).

network is sparse, with typically only a small number of inferred interactions for each health variable.

This inferred causal network can be readily and directly interpreted. For example, we see strong connections between Vitamin-D and self-rated health, between activities of daily living (ADL) score and walking ability, and between glucose and glycated hemoglobin. The sign of the connections indicates the direction of influence. For example, a decrease in gait speed influences an increase in self-reported health score (worse health), an increase in the time required to complete chair rises, and a decrease in grip strength.

Hierarchical clustering on the connection weights is indicated in Fig 4, and the ordering of the variables in the heatmap represents this hierarchy. Many of these inferred clusters of nodes

plausibly fit with known physiology. For example, most blood biomarker measurements (bottom half) are separated from the physical/functional measurements (top half, purple cluster). Other inferred clusters include blood pressure and pulse (orange) and lipids (green).

## Discussion

We have developed a machine learning aging model, DJIN, to predict multidimensional health trajectories and survival given baseline information, and to generate realistic synthetic aging populations—while also learning interpretable network interactions that characterize the dynamics in terms of realistic physiological interactions. The DJIN model uses continuous-valued health variables from the ELSA dataset, including physical, functional, and molecular variables. We have shown that the comprehensive DJIN model performs better than 30 independent regularized linear models that were trained specifically for each separate health variable or survival prediction task.

We were able to further investigate the multi-dimensionality of aging by comparing our DJIN model with a latent-space model that has tunable dimensionality. Accurate death-age predictions require greater than one-dimensional aging models; accurate health trajectories continue to improve as the model dimensionality increases. We conclude that aging health is a high-dimensional process, even for the correlated health variables that we used (see S14 Fig).

Previously, we had built a weighted network model (WNM) using cross-sectional data with only binary health deficits [24]. The WNM did not incorporate continuous health variables and could not be efficiently trained with longitudinal data. As a result, the networks inferred by that model were not robust—and resulted in many qualitatively distinct networks that were all consistent with the observed data. In contrast, the DJIN model uses many continuous valued health variables and can be efficiently trained with large longitudinal datasets. As a result, the DJIN model produces a robust and interpretable interaction network of multi-dimensional aging (see S13 Fig).

Recently, other machine learning models of aging or aging-related disease progression have been emerging [12, 18–20, 44]. Since they each differ significantly in terms of both the datasets, types of data used, and scientific goals, it is still too early to see which approaches are best. We use ELSA data since it is longitudinal (to facilitate modelling trajectories), has many continuous variables (to allow modelling of continuous trajectories and constructing an interaction network that is at the core of our model), and includes mortality (to develop our joint mortality model). The ELSA data is representative of many large-scale aging data sets.

Our scientific goals were to obtain good predictive accuracy from baseline for both health trajectories and mortality, while at the same time obtaining an interpretable network of interactions between health variables [14]. To achieve these goals with the ELSA data, we had to do significant imputation to complete the baseline states. We include stochastic dynamics within a Bayesian model framework to obtain uncertainties for both our predictions and the interaction network. The Bayesian approach is computationally intensive and necessitated a variational approximation to the posterior that tends to underestimate uncertainty [36]. From comparison of observed and synthetic populations (see S9, S10 and S11 Figs), this underestimate appears to be modest. However, there are probably also systemic underestimates of the widths of the posterior distributions of the network weights—and we cannot estimate the scale of that effect in comparison with the observed population.

The DJIN model is not computationally demanding, needing only approximately 12 hours to run with 1 GPU for $M = 25290$ individuals, $B = 19$ background variables, $N = 29$ health-variables, and up to $K = 20$ years of longitudinal data. We expect better predictive performance and generalizability with more individuals $M$. Because of the interactions between health

variables we also expect better predictive performance with more health variables $N$. We note that binary health variables, or mixtures of binary and continuous variables, could be used with only small adjustments. Since computational demands for a forward pass of the model scale approximately linearly with $M$ and $K$, and quadratically with $B + N$, our existing DJIN model is already practical for significantly larger datasets.

In this work we only consider predictions from the baseline state at a single age. We anticipate that individual prediction could be significantly improved by utilizing more than one input time to impute the baseline health state $\mathbf{x}_0$ or by conditioning the predictions on multiple input ages. This conditioning can be done using a recurrent neural network [45, 46]. Observed time-points after baseline can also be used to update the dynamics [47] for predictions of continually observed individuals in personalized medicine applications. However, both of these developments would either require data with more follow-up times than we had available, or limiting predictions to very short time intervals. For these reasons, we chose to model trajectories using only a single baseline health state.

We developed an imputation method that is trained along with the longitudinal dynamics to impute missing baseline data. This imputation method can also be used to generate synthetic individuals conditioned on baseline demographic information. Large synthetic datasets can facilitate the development of future models and techniques by providing high-quality training data [48], and are especially needed given the lack of large longitudinal studies of aging [14]. In S8, S9, S10 and S11 Figs we show that our synthetic population is comparable to the available individuals in the ELSA dataset. We have also provided a synthetic population of nearly $10^7$ individuals with annually sampled trajectories from baseline for 20 years [49].

At the heart of our dynamical model is a directed and signed network that is directly interpretable. The DJIN model does not just make "black-box" predictions, but is learning a plausible physiological model for the dynamics of the health variables. The network is not a correlation/association network (see comparison in S14 Fig) [8, 50, 51], but instead determines how the current value of the health variables influence future changes to connected health variables, leading to coupled dynamics of the health variables. This establishes a predictive link between variables [52]. Similar directed linear networks are inferred in neuroscience with Dynamic Causal Modelling [53, 54]. While previous work on learning networks for discrete stochastic dynamics has been done in the past [55–57], we have used continuous dynamics here. When interpreting the magnitude of weights, links function as in standard regression models: weight magnitudes will be dependent on the variables included in the model, and can decrease if stronger predictor variables are added. Given the efficiency of our computational approach, including more health or background variables is recommended if they are available.

The directed nature of the network connections lend themselves to clinical interpretation— for example ADL impairment has an effect on independent ADL (IADL) impairment and not vice versa, and both have an effect on general function score and vice versa. The directed network of interactions suggests avenues to explore for interventions. For a given intervention (for example drug, exercise, or diet) we can ascertain which effects of the intervention are beneficial and which are deleterious. In principle, we could also predict the outcome of multiple interventions such as in polypharmacy [58]. A similar approach could be taken for chronic diseases or disorders.

While static interventions could simply be included as background variables, our DJIN model could also easily be adapted to allow for time-dependent interventions. These avenues will be increasingly feasible and desirable with longitudinal 'omics data-sets, where the interactions are not already largely determined by previous work. However, we caution that our model does not currently take into account how interventions may change network

interactions over time. We also do not currently account for higher than pair-wise interactions. For example, the interaction between sodium levels, mobility, and diuretics appears to be strong [59], but would not be captured in our current model. Extending our approach to include such effects, and training with data that includes specific time-dependent interventions, is an exciting prospect.

The accuracy of our model can be slightly improved if a network interpretation of the dynamics is not desired—for instance if the goal is only prediction. In Fig 3 and S6 Fig, we show that using a neural network instead of pair-wise network interactions and a high-dimensional latent state can slightly improve health variable prediction accuracy for a handful of variables, but not survival predictions. Our goal here was to demonstrate both good predictions *and* interpretability.

Every aspect of our DIJN model can be made more structured, explicit and "interpretable". The advantage of more interpretable models will be more clearly seen when multiple data-sets are compared—since interpretability facilitates comparisons between cohorts, groups, or even between model organisms. For example, proportional hazards [60] or quadratic hazards [21] could be used for mortality. While these changes would reduce predictive performance compared to our more general DJIN model, they would add interpretability to the survival predictions.

Our work opens the door to many possible follow-up studies. Our DJIN model can be applied to any organism or set of variables that has enough individual longitudinal measurements. With genomic characterization of populations, the background health information $\mathbf{u}_{t_0}$ can be greatly expanded to examine how the intrinsic variability of aging [3, 4] and mortality are affected by genetic variation. By including genomic, lab-test, and functional data we could use the interpretable interactions to determine how different organismal scales of health data interact in determining human aging trajectories. By including drug and behavioral (exercise, diet) interventions as background variables, we can better determine how they affect health during aging. Finally, large longitudinal multi-omics datasets [61, 62] could be used to build integrative models of human health.

We have demonstrated a viable interpretable machine learning (ML) approach to build a model of human aging with a large longitudinal study that can predict health trajectories, generate synthetic individual trajectories, and learn a network of interactions describing the dynamics. The future of these approaches is bright [14], since we are only starting to embrace the complexity of aging with large longitudinal datasets. While ML models can find immediate application in understanding patterns of aging health in populations, we foresee that similar techniques will eventually reach into clinical practice to guide personalized medicine of aging health.

## Methods

### ELSA dataset

We use waves 0–8 of the English Longitudinal study of Aging (ELSA) dataset [25], with 25290 total individuals. We include both original and refreshment samples that joined the study after the start at wave 0. In Table A in S1 Text. we list all variables used. In S1 Fig, we show the number of individuals for which the variable is available at different times from their entrance wave. Each available wave is used as a baseline state for each individual, see section for details.

We extract 29 longitudinally observed continuous or discrete ordinal health variables (treated as continuous) and 19 background health variables (taken as constant with age). We set the gait speed of individuals with values above 4 meters per second to missing, due to a

likely data error. Sporadic missing ages are imputed by assuming the age difference between waves is 2 years—the time difference in the design of the study.

Individuals above age 90 in the ELSA dataset have their age privatized. By assuming the time difference between waves is 2 years, we "deprivatize" these ages within our analysis pipeline. For example, an individual may have recorded ages 87, 89, ⟨*privatized*⟩, ⟨*privatized*⟩, which we deprivatize as 87, 89, 91, 93. When individuals are known to die at an age above 90 at a specific wave, the same approach is done to deprivatize the death age. We have examined the accuracy of reported ages compared to this fixed two-year wave interval deprivatization method (shown in S15 Fig), and we find that the majority of deviations range from 0–1 years (with 78% at 0 years, and an average deviation of 0.23 years)—we expect similar variability for deprivatized ages above 90.

Height is imputed with the last observation carried forward (if it is missing, the first value is carried backwards from the first available measurement). Individuals with no recorded death age are considered censored at their last observed age.

The data is randomly split into separate train (16689 individuals), validation (4173 individuals), and test sets (5216 individuals). The training set is used to train the model, the validation set is used for control of the optimization procedure during training (through a learning rate schedule, see Section below), and the test set is used to evaluate the model after training. Individuals with fewer than 6 variables measured at the baseline age $t_0$ are dropped from the training and validation data. Individuals with fewer than 10 variables measured at the baseline age $t_0$ are dropped from the test data for predictions, while all individuals in the test data are used for population comparisons.

All variables are standardized to mean 0 and standard deviation 1 (computed from the training set); however variables with a skewed age-aggregated distribution $p(\mathbf{y})$ covering multiple orders of magnitude are first log-transformed. Log-scaled variables are indicated in Table A in S1 Text.

## Data augmentation

Since some health variables are measured only at specific visits, using the entrance wave as the only baseline of every individual forces some variables to be rarely observed at baseline, hindering imputation of variables that are only observed at later waves. To mitigate this, we augment the dataset by replicating individuals to use all possible starting points, $t_k^{(m)}, k \in \{0, ..., \mathrm{argmax}_k(t_k^{(m)})\}$. Since individuals have different numbers of observed times we weight individuals in the likelihood who have multiple times available by $s^{(m)} = 1/(\mathrm{argmax}_k(t_k^{(m)}) + 1)$. This helps to prevent the over-weighting of individuals with many possible starting times. Nevertheless, we assume for convenience that replicated individuals are independent in the likelihood. We show a comparison of our model trained with and without this replication in S16 Fig, demonstrating a large improvement in health and survival predictions.

To further augment the available data, we artificially corrupt the input data for training by masking each observed health variable at baseline with probability 0.9. This allows more distinct "individuals" for imputation of the baseline state, and allows us to use self-supervision for these artificially missing values by training to reconstruct the artificially corrupted states.

## DJIN model

We model the temporal dynamics of an individual's health state with continuous-time stochastic dynamics described with stochastic differential equations (SDEs). These SDEs include linear pair-wise interactions between the variables to form a network with a weight matrix $\mathbf{W}$.

We assume the observed health variables $\mathbf{y}_t$ are noisy observations of the underlying latent state variables $\mathbf{x}(t)$, which evolves according to these network SDEs. This allows us to separate measurement noise from the noise intrinsic to the stochastic dynamics of these variables.

These SDEs for $\mathbf{x}(t)$ start from each baseline state $\mathbf{x}_0$, which is imputed from the available observed health state $\mathbf{y}_t$. This imputation process is done using a normalizing-flow variational auto-encoder (VAE) [33]. In this approach, we encode the available baseline information into a latent distribution for each individual, and decode samples from this distribution to perform multiple imputation. The normalizing-flow VAE allows us to flexibly model this latent distribution. The details are described in Section below.

An individual's health state is observed at $K + 1$ times $\{t_k\}_{k=0}^{K}$ with a set of health variables $\{\mathbf{y}_{t_k}\}_{k=0}^{K}$. The vectors $\mathbf{y}_{t_k}$ describe the $N$-dimensional health state of an individual, where each of the $N$ dimensions represents a separate health measurement. Background (demographic, diagnostic, or lifestyle) information observed at baseline, which is described by a $B$-dimensional vector $\mathbf{u}_{t_0}$ used as an auxiliary variable for the dynamics of mortality. We denote the death age or last known age of survival for an individual as $a$, and indicate an individual as censored with $c = 1$ and uncensored with $c = 0$. Our model is described by the following equations:

$$\boldsymbol{\theta} = \{\mathbf{W}, \boldsymbol{\sigma}_\mathbf{y}, \boldsymbol{\sigma}_\mathbf{x}, \boldsymbol{\theta}_\lambda, \boldsymbol{\theta}_p, \boldsymbol{\theta}_f\}. \qquad \text{(Parameters)} \tag{1}$$

$$\mathbf{z}, \boldsymbol{\theta} \sim p(\mathbf{z})p(\boldsymbol{\theta}), \qquad \text{(Prior)} \tag{2}$$

$$\mathbf{x}_0 = \mathbf{o}_{t_0} \odot \mathbf{y}_{t_0} + (1 - \mathbf{o}_{t_0}) \odot \tilde{\mathbf{x}}_0, \ \ \tilde{\mathbf{x}}_0 \sim \mathcal{N}(\mathbf{x}_0 | \boldsymbol{\mu}_\mathbf{x}(\mathbf{z}, \mathbf{u}_{t_0}, t_0; \boldsymbol{\theta}_p), \boldsymbol{\sigma}_\mathbf{y}^2), \qquad \text{(Imputation)} \tag{3}$$

$$dx_i(t) = \left(\sum_{j=1}^{N} W_{ij}x_j(t) + f_i(x_i(t), \mathbf{u}_{t_0}, t; \boldsymbol{\theta}_{f_i})\right)dt + \sigma_{x_i}(\mathbf{x}(t))dB(t), \ \ \mathbf{x}(t_0) = \mathbf{x}_0, \tag{4}$$

$$\mathbf{y}_t \sim \mathcal{N}(\boldsymbol{\psi}^{-1}(\mathbf{x}(t)), \mathrm{diag}(\sigma_\mathbf{y}^2)), \quad \text{(Health observations)} \tag{5}$$

$$S(t) = \exp\left(-\int_{t_0}^{t} \lambda(\{\mathbf{x}(\tau)\}_{\tau \le t'}, \mathbf{u}_{t_0}, t'; \boldsymbol{\theta}_\lambda)dt'\right), \qquad \text{(Survival)} \tag{6}$$

$$a \sim \lambda(\{\mathbf{x}(\tau)\}_{\tau \le a}, \mathbf{u}_{t_0}, a; \boldsymbol{\theta}_\lambda)S(a), \quad \text{(Survival observations)} \tag{7}$$

$$p(\mathbf{z}, \{\mathbf{x}(t)\}_t, \boldsymbol{\theta} | \{\mathbf{y}_{t_k}\}_k, \mathbf{u}_{t_0}, \{\mathbf{o}_{t_k}\}_k, t_0, a, c) \propto p(\boldsymbol{\theta})p(\mathbf{z})p(\mathbf{x}_0 | \mathbf{z}, \mathbf{u}_{t_0}) \times$$

$$p(\{\mathbf{x}(t)\}_t | \mathbf{x}_0, \mathbf{u}_{t_0}, t_0, \boldsymbol{\theta})p(a, c | \{\mathbf{x}(t)\}_t, \mathbf{u}_{t_0}, t_0, \boldsymbol{\theta})\prod_{k=0}^{K} p(\mathbf{y}_{t_k} | \{\mathbf{x}(t_k)\}_k, \mathbf{o}_{t_k}, \boldsymbol{\theta}), \qquad \text{(Inference)} \tag{8}$$

Model parameters ($\theta$) include the explicit network of interactions between health variables ($W$), measurement noise ($\sigma_y$) and dynamical SDE noise ($\sigma_x$), and network weights for mortality RNN ($\theta_\lambda$), imputation VAE decoder ($\theta_p$), and dynamical SDE ($\theta_f$). Eq (2) represents priors on the model parameters and latent state $\mathbf{z}$. We use Laplace($\mathbf{w}|0, 0.05$) priors for the network weights and Gamma($\sigma_\mathbf{y}|1, 25000$) priors for the measurement noise scale parameters. We use a normal (Gaussian) prior distribution for the latent space $\mathbf{z}$. We assume uniform priors for all other parameters.

In Eq (3) we sample the baseline state. The distribution for $\mathbf{x}_0$ given $\mathbf{z}$ is modeled as Gaussian with mean computed from the decoder neural network and the same standard deviation as the measurement noise, $\mathcal{N}(\boldsymbol{\mu}_\mathbf{x}(\mathbf{z}, \mathbf{u}_{t_0}, t_0; \boldsymbol{\theta}_p), \boldsymbol{\sigma}_\mathbf{y}^2)$. The missing value imputation and the dynamics model are trained together simultaneously (see details below). This allows us to utilize the additional longitudinal information for training the imputation method, and helps to avoid an imputed baseline state that leads to poor trajectory or survival predictions. $\odot$ is element-wise multiplication.

Eq (4) describes the SDE network dynamics, starting from the imputed baseline state for each health variable $i = 0, \ldots, N$. We capture single-variable trends with the non-linear $f_i(x_i(t), \mathbf{u}_{t_0}, t; \boldsymbol{\theta}_{f_i})$, and couple the components of $\mathbf{x}(t)$ linearly by the directed interaction matrix $\mathbf{W}$, which represents the strength of interactions between the health variables. In this way, $f_i$ can be thought of as a non-linear function for the diagonal components of this matrix, while $\mathbf{W}$ gives linear pair-wise interactions for the off-diagonal components. The intrinsic diffusive noise in the health trajectories is modeled with Brownian motion with Gaussian increments $d\mathbf{B}(t)$ and strength $\boldsymbol{\sigma}_\mathbf{x}$. The functions $f_i$ and $\boldsymbol{\sigma}_\mathbf{x}$ are parameterized with neural networks.

Eq (5) describes the Gaussian observation model for the observed health state. Measurement noise here is separate from diffusive noise $d\mathbf{B}(t)$ in the SDE from Eq (4). The component-wise transformation $\boldsymbol{\psi}$ applies a log-scaling to skewed variables (indicated in Table A in S1 Text) and z-scores all variables.

Eq (6) describes the survival probability as computed with a recurrent neural network (RNN) for the mortality hazard rate $\lambda$. The RNN allows us to use the stochastic trajectory for the computation of the hazard rate (i.e. it has some memory of health at previous ages), rather than imposing a memory-free process where the hazard rate only depends on the health state at the current age. We use a 2-layer Gated Recurrent Unit (GRU [63]) for the RNN, with hidden state $\mathbf{h}_t$. The initial hidden state $\mathbf{h}_0$ is inferred from the initial health variables $\mathbf{x}(t_0)$, background health information $\mathbf{u}_{t_0}$, and baseline age $t_0$, with a neural network $\mathbf{h}_0 = H(\mathbf{x}(t_0), \mathbf{u}_{t_0}, t_0)$. Eq (7) describes the observation model for survival with age of death or last age known alive $a = \max(t_d, t_c)$, and censoring indicator $c$.

When sampling trajectories from the model, the probability that an individual dies in $[t, t + dt)$ is $\exp(-\lambda(t)\Delta t)$. This is applied at every time-step of the SDE solver to determine specific death times of stochastic realizations of the model.

Instead of just a maximum likelihood point estimate of the network and other parameters of the model, we use a Bayesian approach. This is a natural approach for this model, since the stochastic dynamics of $\mathbf{x}(t)$ are separate from the noisy observations $\mathbf{y}_t$. This also allows us to infer the posterior distribution of the health trajectories and interaction network, and so lets us estimate the robustness of the inferred network and the distribution of possible predicted trajectories, given the observed data. In Eq (8) we show the form of the unnormalized posterior distribution.

## Variational approximation for scalable Bayesian inference

While sampling based methods of inference for SDE models do exist [64, 65], these are generally not scalable to large datasets or to models with many parameters. Instead, we use an approximate variational inference approach [66, 67]. We assume a parametric form of the posterior that is optimized to be close to the true posterior. While variational approximations tend to underestimate the width of posterior distributions and simplify correlations, they typically capture the mean [36]. For the rest of the methods we denote posterior approximations as $q(.)$, and prior distributions, likelihood distributions, and the true posterior with $p(.)$.

Our factorized variational approximation to the posterior in Eq (8) is

$$q(\mathbf{z}, \mathbf{x}(t), \boldsymbol{\theta}|\mathbf{y}_0, \mathbf{u}_{t_0}, \mathbf{o}_{t_0}, t_0, \boldsymbol{\phi}) = q(\mathbf{z}|\mathbf{y}_0, \mathbf{u}_{t_0}, \mathbf{o}_{t_0}, t_0, \boldsymbol{\phi}_z)q(\mathbf{x}(t)|\mathbf{x}_0, \mathbf{u}_{t_0}, t, \boldsymbol{\phi}_x)q(\boldsymbol{\theta}|\boldsymbol{\phi}_\theta),$$
$$\{\mathbf{x}(t)\}_t \sim q(\mathbf{x}(t)|\mathbf{x}_0, \mathbf{u}_{t_0}, t, \boldsymbol{\phi}_x) \Rightarrow \tag{9}$$

$$d\mathbf{x}(t) = (\bar{\mathbf{W}}\mathbf{x} + \mathbf{f}(\mathbf{x}, \mathbf{u}_{t_0}, t; \boldsymbol{\theta}_f) + \mathbf{g}(\mathbf{x}, \mathbf{u}_{t_0}, t; \phi))dt + \boldsymbol{\sigma}_x(\mathbf{x}(t))d\mathbf{B}(t), \tag{10}$$

with variational parameters $\phi = \{\phi_x, \phi_z, \phi_\theta\}$. Instead of assuming an explicit parametric form for $q(\mathbf{x}(t)|\phi_x)$, we instead assume the trajectories $\{\mathbf{x}(t)\}_t$ are described by samples from a posterior SDE with drift modified by including a small fully connected neural network $\mathbf{g}$ [68]. This approach allows an efficient and flexible form of the variational posterior in Eq 9. $\bar{\mathbf{W}}$ is the posterior mean of the network weights. The functional form of the posterior drift is both more general and more easily trainable than the network SDE in Eq 4, but ultimately is forced to be close to the network dynamics in Eq (4) by the loss function. The loss function for this approach has been previously derived [66, 67]. The imputed baseline states $\mathbf{x}_0$ are averaged over.

For the latent state $\mathbf{z}$, the approximate posterior takes the form

$$\boldsymbol{\mu}_z, \ \boldsymbol{\sigma}_z, \ \boldsymbol{\gamma}_z \ = \ \text{Encoder}(\tilde{\mathbf{y}}_{t_0}, \mathbf{o}_{t_0}, \mathbf{u}_{t_0}, t_0, \phi_z), \tag{11}$$

$$\tilde{\mathbf{y}}_{t_0} \ = \ \mathbf{o}_{t_0} \odot \mathbf{y}_{t_0} + (1 - \mathbf{o}_{t_0}) \odot \epsilon_{\mathbf{y}_{s,t_0},\text{pop}}, \tag{12}$$

$$q(\mathbf{z}|\mathbf{y}_{t_0}, \mathbf{u}_{t_0}, \mathbf{o}_{t_0}, t_0, \boldsymbol{\phi}_z) \ \equiv \ q(\mathbf{z}^{(L)}|\tilde{\mathbf{y}}_{t_0}, \mathbf{u}_{t_0}, \mathbf{o}_{t_0}, t_0, \boldsymbol{\phi}_z) \tag{13}$$

$$= \ \mathcal{N}(\mathbf{z}^{(0)}|\boldsymbol{\mu}_z, \boldsymbol{\sigma}_z^2)\prod_{l=1}^{L}\left|\det\frac{\partial a^{(l)}(\mathbf{z}^{(l)}, \boldsymbol{\gamma}_z, \phi_z)}{\partial \mathbf{z}^{(l)}}\right|^{-1}, \tag{14}$$

where the functions $a^{(l)}$ are RealNVP normalizing flows [69] used to transform the Gaussian base distribution for $\mathbf{z}^{(0)}$ to the non-Gaussian posterior approximation, conditioned on the specific individual with $\boldsymbol{\gamma}_z$. These are invertible neural networks that transform probability distributions while maintaining normalization. We use $L = 3$ normalizing flow networks. In Eq 12 we fill in missing values in the observed health state since $\mathbf{o}$ is a mask of observed variables and $\epsilon_{\mathbf{y}_{s,t_0},\text{pop}}$ is sampled from a Gaussian distribution with the sex and age-dependent sample mean and standard deviation. $\odot$ is element-wise multiplication. These filled in values are temporarily input to the encoder neural network, and replaced after imputation.

The variational parameters $\boldsymbol{\phi}$ of the approximate posterior are optimized to minimize the KL divergence between the approximation and the true posterior. This minimized KL divergence provides a lower bound to the model evidence that can be maximized,

$$\log p(\{\mathbf{y}_{t_k}\}_k|\mathbf{u}_{t_0}, \mathbf{o}_{t_0}, t_0) \geq \mathbb{E}_{\boldsymbol{\theta}, \mathbf{z}, \mathbf{x}_0|\mathbf{z}, \{\mathbf{x}(t)\}_t|\mathbf{x}_0}\Big[$$
$$\log \{p(\boldsymbol{\theta})p(\mathbf{z})p(\{\mathbf{x}(t)\}_t|\mathbf{x}_0, \mathbf{u}_{t_0}, \boldsymbol{\theta})p(a, c|\{\mathbf{x}(t)\}_t, \mathbf{u}_{t_0}, t_0)\prod_k p(\mathbf{y}_{t_k}|\mathbf{x}(t_k), \mathbf{o}_{t_k}, \boldsymbol{\theta})\} \tag{15}$$
$$-\log \{q(\mathbf{z}|\mathbf{y}_0, \mathbf{u}_{t_0}, \mathbf{o}_{t_0}, t_0)q(\boldsymbol{\theta})q(\{\mathbf{x}(t)\}_t|\mathbf{x}_0, \mathbf{u}_{t_0})\}\Big],$$

where in the expectation $\boldsymbol{\theta}$, $\mathbf{z}$, and $\{\mathbf{x}(t)\}_t$ are sampled from their respective posterior

distributions. The imputed baseline state is sampled as,

$$\boldsymbol{\mu}_{\mathbf{x}} \quad = \quad \text{Decoder}(\mathbf{z}, \mathbf{u}_{t_0}, t_0) \tag{16}$$

$$\tilde{\mathbf{x}}_0 \quad \sim \quad \mathcal{N}(\boldsymbol{\mu}_{\mathbf{x}}, \boldsymbol{\sigma}_{\mathbf{y}}^2) \tag{17}$$

$$\mathbf{x}_0 \quad = \quad \mathbf{o}_{t_0} \odot \mathbf{y}_{t_0} + (1 - \mathbf{o}_{t_0}) \odot \tilde{\mathbf{x}}_0. \tag{18}$$

Note that we keep the observed value $\mathbf{y}_{t_0}$ when available.

The final objective function to be maximized is $\mathcal{L}$, where the derivation is provided in the S1 Text. We obtain

$$
\begin{aligned}
\mathcal{L}(\phi) \quad = \quad & \mathbb{E}\Bigg[ \sum_{k=0}^{K} \mathbf{o}_{t_k} \odot \log \mathcal{N}(\mathbf{y}_{t_k} | \mathbf{x}(t_k), \boldsymbol{\sigma}_{\mathbf{y}}) \\
& + \quad (1-c)[\log \lambda(a|\mathbf{x}(t), \mathbf{u}_{t_0}, t_0) + \log S(a|\mathbf{x}(t), \mathbf{u}_{t_0}, t_0)] \\
& + \quad \int_{t_0}^{a} c \log S(t|\mathbf{x}(t), \mathbf{u}_{t_0}, t_0) dt + \int_{a}^{a_{\max}} (1-c) \log(1 - S(t|\mathbf{x}(t), \mathbf{u}_{t_0}, t_0)) dt \\
& - \quad \frac{1}{2} \int_{t_0}^{a} ||\boldsymbol{\sigma}_x^{-1} \odot \left( \mathbf{W}\mathbf{x} - \bar{\mathbf{W}}\mathbf{x} - \mathbf{g}(\mathbf{x}, \mathbf{u}_{t_0}, t) \right)||_2^2 dt \Bigg] \\
& - \quad KL(q(\boldsymbol{\theta})||p(\boldsymbol{\theta})) - KL(q(\mathbf{z}^{(0)}|\mathbf{y}_0, \mathbf{u}_{t_0}, \mathbf{o}_{t_0}, t_0)||p(\mathbf{z}^{(0)})) \\
& + \quad \sum_{l=1}^{L} \log \left| \det \frac{\partial a^{(l)}(\mathbf{z}^{(l)}, \boldsymbol{\gamma}_z, \phi_z)}{\partial \mathbf{z}^{(l)}} \right|,
\end{aligned}
\tag{19}
$$

as the loss function for each individual. This is for all individuals in the data multiplied by the sample weights $s^{(m)}$ for each individual $m$. The first 3 lines of this loss are the likelihood for the data, including both health and survival. We penalize the survival probability by integrating the probability of being dead from the death age $a$ to $a_{\max}$, which better estimates survival probabilities [70]. We set $a_{\max} = 5$ years. Otherwise, it is difficult for the model to learn $S \rightarrow 0$ for large $t$. The last 3 lines are the KL-divergence terms for variational inference. The very last term is for the normalizing flow portion of the variational auto-encoder.

To simplify the evaluation of $\mathcal{L}$ and decrease the number of parameters, we assume independent Gamma posteriors for each measurement error parameter $\sigma_y$ with separate shape $\alpha_i$ and rate $\beta_i$. We also assume independent Laplace posteriors for each of the network weights $W_{ij}$ with separate means $\bar{W}_{ij}$ and scales $b_{ij}$. For the approximate distribution of all other parameters we use delta functions, and together with uniform priors this leads to simplifying the approach to just optimizing these parameters instead of optimizing variational parameters of the posterior.

## Summarized training procedure

1. Pre-process data. Assign $N$ dynamical health variables and $B$ static health variables. Reserve validation and test data from training data.

2. Sample batch and apply masking corruption and temporarily fill in missing values with samples from the population distribution,

$$\tilde{\mathbf{y}}_{t_0} = \mathbf{c} \odot \mathbf{o}_{t_0} \odot \mathbf{y}_{t_0} + (1 - \mathbf{c} \odot \mathbf{o}_{t_0}) \odot + \epsilon_{\mathbf{y}_{s,t_0},\text{pop}}, \tag{20}$$

$$\mathbf{c} \sim \text{Bernoulli}(0.9). \tag{21}$$

3. Impute initial state $\mathbf{x}_0$ with the VAE and compute the initial memory state of the mortality rate GRU,

$$\mathbf{z} \quad \sim \quad q(\mathbf{z}|\tilde{\mathbf{y}}_{t_0}, \mathbf{u}_{t_0}, \mathbf{c} \odot \mathbf{o}_{t_0}, t_0), \tag{22}$$

$$\tilde{\mathbf{x}}_0 \quad \sim \quad \mathcal{N}(\mathbf{x}_0|\boldsymbol{\mu}_{\mathbf{x}}(\mathbf{z}, \mathbf{u}_{t_0}, t_0), \boldsymbol{\sigma}_{\mathbf{y}}^2) \tag{23}$$

$$\mathbf{x}_0 \quad = \quad \mathbf{o}_{t_0} \odot \mathbf{y}_{t_0} + (1 - \mathbf{o}_{t_0}) \odot \tilde{\mathbf{x}}_0, \tag{24}$$

$$\mathbf{h}_{t_0} \quad = \quad \mathbf{H}(\mathbf{x}_0, \mathbf{u}_{t_0}, t_0). \tag{25}$$

4. Sample trajectory from the SDE solver for the posterior SDE and compute mortality rate from GRU,

$$\{\mathbf{x}(t)\}_t \quad = \quad \text{SDESolver}(\mathbf{x}_0, \mathbf{u}_{t_0}, t_0), \tag{26}$$

$$\{S(t)\}_t \quad = \quad \text{GRU}(\{\mathbf{x(t)}\}_t|\mathbf{h}_{t_0}). \tag{27}$$

5. Compute the gradient of the objective function (Eq 19) and update parameters, returning to step 2 until training is complete.

6. Evaluate model performance on test data.

## Network architecture and Hyperparameters

The different neural networks used are summarized in Table C in S1 Text. We use ELU activation functions for most hidden layer non-linearities, unless specified otherwise. We have $N = 29$ dynamical health variables, and $B = 19$ static health variables. Additionally, we append a mask to the static health variables indicating which are missing, of size 17 (sex and ethnicity are never missing).

The functions $f_i$ in Eq (4) are feed-forward neural networks with input size $2 + B + 17$, hidden layer size 12, and output size 1. Each $f_i$, $i \in \{1, \ldots, N\}$ has its own weights. The noise function $\boldsymbol{\sigma}_{\mathbf{x}}$ has input size $N$, hidden layer size $N$, and output size $N$. The posterior drift $\mathbf{g}$ is a fully-connected feed-forward neural network with input size $N + B + 1 + 17$, hidden layer size 8, and output size $N$. The VAE encoder has input size $2N + B + 1 + 17$, hidden layer sizes 95 and 70, and output size 40, with batch normalization applied before the activation functions for each hidden layer. The VAE decoder has input size $20 + B + 17$, hidden layer size 65, and output size $N$ with batch normalization applied before the activation for the hidden layer. The size

of the latent state $\mathbf{z}$ is 20. The mortality rate $\lambda$ is a 2-layer GRU [63] with a hidden layer sizes of 25 and 10.

We use $L = 3$ normalizing flow networks to transform the latent distribution from the Gaussian $\mathbf{z}^{(0)}$ to $\mathbf{z}$. We use RealNVP normalizing flow networks [69] with layer sizes 30, 24, and 10 with batch normalization before a Tanh activation function for the hidden layer. The size of $\gamma_z$ is 10.

We use batchsize of 1000 and learning rate $10^{-2}$ with the ADAM optimizer [71]. We decay the learning rate by a factor of 0.5 at loss plateaus lasting for 40 or more epochs. We use KL-annealing with $\beta$ increasing linearly from 0 to 1 during the first 300 epochs for the KL loss terms for $q(\mathbf{x}(t))$ and $q(\mathbf{z}(t))$, and increase linearly from 0 to 1 from 300 to 500 epochs for the KL terms for the prior on $\mathbf{W}$. SDEs are solved with the strong order 1.0 stochastic Runge-Kutta method [72] with a constant time-step of 0.5 years. Integrals in the likelihood are computed with the trapezoid method using the same discretization as the dynamics.

## Latent space models

We compare our pair-wise interactions network model with latent space models, where we directly incorporate dynamics for the latent state $\mathbf{z}(t)$ and apply the decoder to estimate the health variables $\mathbf{x}(t)$ at specific ages. With this approach we do not need to impute the baseline state of health variables, or to directly include dynamics for the observed health state. Rather an encoder maps the baseline health state $\mathbf{y}_{t_0}$ to the baseline latent state $\mathbf{z}_0$, dynamics are run on this latent space for $\mathbf{z}(t)$, and a decoder directly maps the latent states $\mathbf{z}(t)$ to the predicted output of the health variables $\mathbf{y}_t$. In this model, we also can choose the size of the latent state $\mathbf{z}$, and so we use this approach to explore how many dimensions are required for good predictions of health outcomes and survival.

These models have the form,

$$\mathbf{z}_0, \boldsymbol{\theta} \sim p(\mathbf{z}_0)p(\boldsymbol{\theta}) \qquad \text{(Prior)}$$

$$d\mathbf{z}(t) = \mathbf{f}(\mathbf{z}(t), \mathbf{u}_{t_0}, t; \boldsymbol{\theta}_f)dt + \boldsymbol{\sigma}_\mathbf{z}(\mathbf{z}(t))d\mathbf{B}(t), \ \mathbf{z}(t_0) = \mathbf{z}_0, \qquad \text{(Dynamics)}$$

$$S(t) = \exp(-\int_{t_0}^t \lambda(\{\mathbf{z}(\tau)\}_{\tau \le t'}, \mathbf{u}_{t_0}, t'; \boldsymbol{\theta}_\lambda)dt'), \qquad \text{(Survival)}$$

$$\mathbf{y}_t \sim \mathcal{N}\left(\boldsymbol{\psi}^{-1}(\boldsymbol{\mu}(\mathbf{z}(t), \mathbf{u}_{t_0}; \boldsymbol{\theta}_p)), \text{diag}(\sigma_\mathbf{y}^2)\right), \qquad \text{(Health observation)}$$

$$a \sim \lambda(\{\mathbf{z}(\tau)\}_{\tau \le a}, \mathbf{u}_{t_0}, a; \boldsymbol{\theta}_\lambda)S(a), \qquad \text{(Survival observation)}$$

$$p(\{\mathbf{z}(t)\}_t, \boldsymbol{\theta}|\{\mathbf{y}_{t_k}\}_k, \mathbf{u}_{t_0}, t_0, a, c) \propto p(\boldsymbol{\theta})p(\mathbf{z}_0)p(\{\mathbf{z}(t)\}_t|\mathbf{z}_0, \mathbf{u}_{t_0}, t, \boldsymbol{\theta}) \times$$
$$p(a, c|\{\mathbf{z}(t)\}_t, \mathbf{u}_{t_0}, t, \boldsymbol{\theta})\prod_k p(\mathbf{y}_{t_k}|\{\mathbf{z}(t_k)\}_k, \boldsymbol{\theta}), \qquad \text{(Inference)}$$

$$\boldsymbol{\theta} = \{\mathbf{W}, \sigma_\mathbf{y}, \sigma_\mathbf{x}, \boldsymbol{\theta}_\lambda, \boldsymbol{\theta}_p, \boldsymbol{\theta}_f\}, \qquad \text{(Parameters)}$$

where instead of the variable-wise neural networks in the pair-wise network model, the function $\mathbf{f}$ is now a full feed-forward neural network including the interactions between all variables. The function $\boldsymbol{\mu}$ is a decoder neural network which outputs the mean of a Gaussian distribution for the health variables $\mathbf{y}_t$, from the latent state at that age.

To create a 1D summary model that includes all information in $\mathbf{u}_{t_0}$ in the 1-dimensional latent state $z$, we use this same model but remove all instances of $\mathbf{u}_{t_0}$ (except sex, ethnicity, and country of birth components) from every function except the encoder.

Other than the size of the latent state $\mathbf{z}$, all other hyperparameters and the training procedure remain the same as the DJIN model described above. In particular, the form of the loss function remains the same, except that the priors for $\mathbf{W}$ are removed, and the form of the drift function in the SDE is adjusted. The parameters for these alternative models are trained with the loss function using the same approach as our primary DJIN model.

## Evaluation metrics

**RMSE scores.** Longitudinal health trajectory predictions are assessed with the Root-Mean-Square Error (RMSE) of the predictions with respect to the observed values. The RMSE is evaluated for each health variable and is weighted by the sample weights $s^{(m)}$. We compute these RMSE values for predictions for a specific age $t_k$ for variable $i$,

$$\text{RMSE}_i(t_k) = \sqrt{\frac{1}{M}\sum_{m=1}^{M}s^{(m)}(\psi_i^{-1}(x_i^{(m)}(t_k)) - y_{i,t_k}^{(m)})^2}, \qquad (28)$$

where the inverse transform $\psi_i^{-1}$ reverse any log-scaling and the z-scoring performed on the variables. The index $(m)$ indicates the individual, for $M$ total individuals.

**Time-dependent C-index.** The C-index measures the probability that the model correctly identifies which of a pair of individuals live longer. Our model contains complex time-dependent effects where survival curves can potentially intersect, so we use a time-dependent C-index [37],

$$
\begin{aligned}
C_{\text{td}} &= \Pr(\hat{S}^{(m_1)}(a^{(m_1)}) < \hat{S}^{(m_2)}(a^{(m_1)})|a^{(m_1)} < a^{(m_2)}, c^{(m_1)} = 0) \\
&= \frac{\sum_{m_1,m_2}s^{(m_1)}s^{(m_2)}\delta[\hat{S}^{(m_1)}(a^{(m_1)}) < \hat{S}^{(m_2)}(a^{(m_1)})]\delta[a^{(m_1)} < a^{(m_2)}]\delta[c^{(m_1)} = 0]}{\sum_{m_1,m_2}s^{(m_1)}s^{(m_2)}\delta[a^{(m_1)} < a^{(m_2)}]\delta[c^{(m_1)} = 0]},
\end{aligned}
\qquad (29)
$$

where $s^{(m)}$ are individual sample weights. We denote death ages by $t_d$ and censoring ages by $t_c$, and define $a^{(m)} = \min(t_c^{(m)}, t_d^{(m)})$ as the last observed age for censored individuals ($c^{(m)} = 1$) or the death age for uncensored individuals ($c^{(m)} = 0$). The indexes $(m_1)$ and $(m_2)$ indicate the pair of individuals that are being compared. Delta functions $\delta[]$ have value 1 if the argument is true, otherwise have value 0.

**Brier score.** The Brier score compares predicted individual survival probabilities to the exact survival curves, i.e. a step function where $S = 1$ while the individual is alive, and $S = 0$ when the individual is dead. The censoring survival function $G(t)$ is computed from the Kaplan-Meier estimate of the censoring distribution (using censoring as events rather than the death [40]), which is used to weight the individuals to account for censoring. Then the Brier score is computed for all possible death ages,

$$\text{BS}(t) = \frac{1}{M}\sum_m s^{(m)}\left[\frac{\delta[a^{(m)} \leq t, c^{(m)} = 0](S^{(m)}(t))^2}{G(a^{(m)})} + \frac{\delta[a^{(m)} > t](1 - S^{(m)}(t))}{G(t)}\right]. \qquad (30)$$

Each individual is indexed $(m)$. Delta functions $\delta[]$ have value 1 if the argument is true, otherwise have value 0.

**D-calibration.** For well-calibrated survival probability predictions, we expect $p\%$ of individuals to have survived past the $p$th quantile of the survival distribution. This can be evaluated using D-calibration, and we follow the previously developed procedure [41] for computing the

D-calibration statistic. The result is a discrete distribution that should match a uniform distribution if the calibration is perfect.

We use a $\chi^2$ test to compare to the uniform distribution. Using 10 bins, we use a $\chi^2$ test with 9 degrees of freedom. Larger p-values (and smaller $\chi$ scores) indicate that the survival probabilities are more uniformly distributed, as desired.

**2-sample classification tests.** To assess the quality of our synthetic population, we train a logistic regression classifier and evaluate its ability to differentiate between the observed and synthetic populations [18, 19, 43, 44]. Ideally, a synthetic population would be indistinguishable from the observed population, giving a classification accuracy of 50%.

Our classifier takes the current age $t$, the synthetic or observed health variables $\mathbf{y}_t$, and the background health information variables $\mathbf{u}_{t_0}$, and then outputs the probability of being a synthetic individual or a real observed individual from the data-set. Missing values in the observed population are imputed with the sex and age-dependent sample mean, and these same values are applied to the synthetic health trajectories by masking the predicted values.

**Hierarchical clustering.** We perform hierarchical clustering on the network weights $\mathbf{W}$. This is done by constructing a dissimilarity matrix,

$$\boldsymbol{\omega} \quad = \quad (\mathbf{W}^T + \mathbf{W})/2, \tag{31}$$

$$\mathbf{D} \quad = \quad \max(\boldsymbol{\omega}) - \boldsymbol{\omega}, \tag{32}$$

and then using this dissimilarity matrix $\mathbf{D}$ to perform agglomerative clustering with the average linkage [73]. We use the Scikit-learn [74] package.

## Comparison with linear models

**Imputation for comparison models.** For the linear survival and longitudinal models, we use MICE for imputation [38] with a random forest model [39]. We impute with the mean of the estimated values. We use 40 trees and do a hyperparameter search over the maximum tree depth. We use the Scikit-learn [74] package.

**Proportional hazards survival model.** To compare with a suitable baseline model for survival predictions, we use a proportional hazards model [60] with the Breslow baseline hazard estimator [75]:

$$\lambda(t|t_0, \mathbf{y}_{t_0}, \mathbf{u}_{t_0}) \quad = \quad \exp(\beta_0 t_0 + \boldsymbol{\beta}_y \cdot \mathbf{y}_{t_0} + \boldsymbol{\beta}_u \cdot \mathbf{u}_{t_0}), \tag{33}$$

$$S(t|t_0, \mathbf{y}_{t_0}, \mathbf{u}_{t_0}) \quad = \quad \exp(-\hat{\Lambda}_0^{\mathrm{Br}}(t)\lambda(t|t_0, \mathbf{y}_{t_0}, \mathbf{u}_{t_0})). \tag{34}$$

We include elastic net regularization [76] for the coefficients of the covariates.

**Linear trajectory model.** We use a simple linear model for health trajectories given baseline data,

$$y_{t_k,i} \quad = \quad y_{t_0,i} + \beta(\mathbf{y}_{t_0}, \mathbf{u}_{t_0}, t_0)(t_k - t_0), \tag{35}$$

$$\beta_i(\mathbf{y}_{t_0}, \mathbf{u}_{t_0}, t_0) \quad = \quad \beta_{0,i} t_0 + \boldsymbol{\beta}_{1,i} \cdot \mathbf{y}_{t_0} + \boldsymbol{\beta}_{2,i} \cdot \mathbf{u}_{t_0}, \tag{36}$$

trained independently for each variable $i$. The parameters $\beta_{0,i}$, $\boldsymbol{\beta}_{1,i}$, and $\boldsymbol{\beta}_{2,i}$ are trained with elastic net regularization.

**Linear models' hyperparameters.** We perform a random search over the $L_1$ and $L_2$ elastic net regularization parameters and the MICE random forest maximum depth using the validation set. The regularization term in the elastic net models is $\alpha l_{1,ratio}||\beta||_1 + \frac{1}{2}\alpha(1 - l_{1,ratio})||\beta||_2^2$,

the common form of elastic net regularization used in Scikit-learn [74], the package we use to implement the elastic net linear model. We do the random search over $\log_{10} \alpha \in [-4, 0]$, $\log_{10} l_{1,ratio} \in [-2, 0]$, and maximum tree depth in [5, 10] for 25 iterations.

   We find the parameters $\alpha = 0.40423$, $l_{1,ratio} = 0.55942$, and a maximum tree depth of 10 for the longitudinal model hyperparameters. We find the parameters $\alpha = 0.00016$, $l_{1,ratio} = 0.15613$, and a maximum tree depth of 10 for the survival model hyperparameters.

## Supporting information

**S1 Text. Supplemental information.** Supplemental derivations, alternate models, and tables. Table A. Variables used from the ELSA dataset. Background variables are only used at the first time-step, as $\mathbf{u}_{t_0}$. Longitudinal variables are predicted in $\mathbf{y}_t$. All variables are z-scored; additional transformations before z-scoring are indicated. Table B. Activities of daily living (ADL) and Instrumental activities of daily living (IADL) from the ELSA dataset, for a total of 10 ADL and 13 IADL. Table C. Neural network architectures used in the DJIN model, as described in Fig 1 and "Network architecture and Hyperparameters" of the methods. The health variables $\mathbf{y}_{t_0}$ are size $N = 29$, the health variable observed mask $\mathbf{o}_{t_0}$ is of size $N = 29$, and the background health variables $\mathbf{u}_{t_0}$ with appended missing mask are of size $B + 17 = 36$.
(PDF)

**S1 Fig. Coverage of ELSA dataset.** Number of individuals with measurements vs number of years after entrance to the study. Although ELSA study design has each wave 2 years apart, the age of individuals can change between 1 and 3 years between visits. Health variables (purple shading) are included in $\mathbf{y}_t$. Background variables (green shading) are included in $\mathbf{u}_{t_0}$. Indicated at the bottom (orange shading) are the number of deaths reported, the number of individuals, and the average coverage percentage for those individuals. The darker shading indicates more measurements, relative to the maximum for that variable.
(TIF)

**S2 Fig. Feed-forward mortality rate model. a)** Time-dependent C-index stratified vs age (points) and for all ages (line). Results are shown for the feed-mortality mortality rate model (purple), the DJIN network model with a recurrent neural network mortality rate shown in the main results (red) and a Elastic net Cox model (green). (Higher scores are better). **b)** Brier scores for the survival function vs death age. Integrated Brier scores (IBS) over the full range of death ages is also shown. The Breslow estimator is used for the baseline hazard in the Cox model (Cox-Br). (Lower scores are better). Our DJIN model performs better than the feed-forward mortality model. **c)** RMSE scores when the baseline value is observed for each health variable for predictions at least 5 years from baseline, scaled by the RMSE score from the age and sex dependent sample mean (relative RMSE scores). We show the predictions from the feed-forward model starting from the baseline value (purple stars), our DJIN model (red circles), predictions assuming a static baseline value (blue diamonds), an elastic-net linear model (green squares). (Lower is better). **d)** Relative RMSE scores when the when the baseline value for each health variable is imputed for predictions past 5 years from baseline. We show the predictions from the feed-forward mortality model starting from the imputed value (purple stars), our DJIN model (red circles), and predictions with an elastic-net linear model (green squares). For longitudinal predictions, the DJIN model is almost equivalent to the feed-forward mortality model.
(TIF)

**S3 Fig. D-calibration comparison with elastic-net Cox model. a)** D-calibration of survival predictions for the DJIN model. Estimated survival probabilities are expected to be uniformly distributed (dashed black line). We use Pearson's $\chi^2$ test to assess the distribution of survival probabilities finding $\chi^2 = 1.3$ and $p = 1.0$ and an elastic net Cox model. (Higher p-values and smaller $\chi^2$ statistics are better). **b)** D-calibration of survival predictions for the elastic-net Cox model. Estimated survival probabilities are expected to be uniformly distributed (dashed black line). We use Pearson's $\chi^2$ test to assess the distribution of survival probabilities finding $\chi^2 = 2.1$ and $p = 1.0$. Error bars show the standard deviation.
(TIF)

**S4 Fig. Longitudinal predictions vs proportion missing.** Relative RMSE up to six-years past baseline for longitudinal predictions of each health variable plotted against the proportion of observations where that variable was missing. Red circles show our network DJIN model, while green squares show the elastic net linear model. Predictions degrade only at high missingness.
(TIF)

**S5 Fig. Model example trajectories.** We show example predictions for 3 test individuals (**a**, **b**, and **c**). For each individual we show the top 6 best predicted health variables from Fig 2 in the main results. Black circles show the observed ELSA data. Red lines indicate the mean predicted $\mathbf{x}(t)$ and the red shaded region is one standard deviation from the predicted mean trajectory. Green lines indicate the linear model prediction (which appear curved for log-scaled variables such as ferritin). The average relative RMSE for each variable for each individual is shown.
(TIF)

**S6 Fig. 30-dimensional latent variable model with full neural network drift. a)** Time-dependent C-index stratified vs age (points) and for all ages (line). Results are shown for the full neural network model (purple), the DJIN network model shown in the main results (red) and a Elastic net Cox model (green). (Higher scores are better). **b)** Brier scores for the survival function vs death age. Integrated Brier scores (IBS) over the full range of death ages is also shown. The Breslow estimator is used for the baseline hazard in the Cox model (Cox-Br). (Lower scores are better). **c)** RMSE scores when the baseline value is observed for each health variable for predictions at least 5 years from baseline, scaled by the RMSE score from the age and sex dependent sample mean (relative RMSE scores). We show the predictions from the full neural network model starting from the baseline value (purple stars), our network model (red circles), predictions with a static baseline value (blue diamonds), an elastic-net linear model (green squares). (Lower is better). **d)** Relative RMSE scores when the when the baseline value for each health variable is imputed for predictions past 5 years from baseline. We show the predictions from the full neural network model starting from the imputed value (purple stars), our network model (red circles), and predictions with an elastic-net linear model (green squares).
(TIF)

**S7 Fig. One-dimensional summary model. a)** Time-dependent C-index stratified vs age (points) and for all ages (line). Results are shown for the 1D summary model (purple), the DJIN network model shown in the main results (red) and a Elastic net Cox model (green). (Higher scores are better). **b)** Brier scores for the survival function vs death age. Integrated Brier scores (IBS) over the full range of death ages is also shown. The Breslow estimator is used for the baseline hazard in the Cox model (Cox-Br). (Lower scores are better). **c)** RMSE scores when the baseline value is observed for each health variable for predictions at least 5 years from baseline, scaled by the RMSE score from the age and sex dependent sample mean (relative RMSE scores). We show the predictions from the 1D summary model starting from the

baseline value (purple stars), our network model (red circles), predictions assuming a static baseline value (blue diamonds), an elastic-net linear model (green squares). (Lower is better). **d)** Relative RMSE scores when the when the baseline value for each health variable is imputed for predictions past 5 years from baseline. We show the predictions from the 1D summary model starting from the imputed value (purple stars), our network model (red circles), and predictions with an elastic-net linear model (green squares).
(TIF)

**S8 Fig. Synthetic population classification.** We use a logistic regression classifier to evaluate the quality of our generated synthetic population by the classifier's ability to differentiate the synthetic population from the observed sample. The boxplot shows the median with the horizontal lines, interquartile range with the box, and 1.5x from the interquartile range with the whiskers. Completely indistinguishable natural and synthetic populations would have a classification accuracy of 0.5. We show the classification accuracy vs years from baseline, showing low classification accuracies that increase slowly with time from baseline in the DJIN model, and the DJIN model is equivalent or better than non-linear latent variable models.
(TIF)

**S9 Fig. Synthetic population baseline distributions.** Each plot shows a synthetic baseline marginal distribution (red shading) for each variable. The synthetic baseline is generated given the background variables $\mathbf{u}_{t_0}$ for the test set. Also shown is the observed distribution (blue shading). Log-scaled variables are shown with a logarithmic x-axis, and are indicated with an *.
(TIF)

**S10 Fig. Synthetic population trajectories.** Red lines show the synthetic population trajectory marginal distribution means for each variable. Red shaded regions indicate 1 standard deviation away from the mean. Synthetic trajectories are generated from the baseline states shown in S9 Fig. Blue lines and shaded regions indicate the corresponding means and 1 standard deviation away for the observed population.
(TIF)

**S11 Fig. Synthetic survival distribution.** Survival curve for synthetic and observed populations, as indicated. The shaded regions show the 95% confidence intervals for Kaplan-Meier curves. The observed sample censoring distribution is applied to the synthetic population. The survival probability is approximately the same until 90 years, indicating that the mortality of the synthetic population is representative until older ages.
(TIF)

**S12 Fig. Network interaction criterion.** Criteria for determining robust connections. We show the posterior mean of the network weights $\{W_{ij}\}$ vs. the proportion of the posterior above zero for weights with a positive mean, and below zero for weights with a negative mean. The vertical dashed red line shows the criteria for robust connections (which are used in Fig 4), which is a 99% credible interval around the mean not containing zero. We see that larger weights are all credible, while many smaller weights are not.
(TIF)

**S13 Fig. Network robustness.** Inferred network for 4 different fits of the model. These networks visually look very similar, however there are some differences in magnitudes of the connections.
(TIF)

**S14 Fig. Comparison with correlation network. a)** Pearson correlation network between the health variables for all individuals at all time-points, values are pruned for p-values above 0.01. **b)** Our model interaction network, for comparison. Weights are pruned when the 99% posterior credible interval includes zero.
(TIF)

**S15 Fig. Testing deprivatization of ELSA ages.** All known unprivatized ages past the first age for each individual in the ELSA dataset are set to missing. This allows us to test our approach to deprivatizing ages above 90, using the known ages below 90. **a)** Scatter plot of observed age vs deprivatized age. The green dashed light highlights perfect deprivatization. All errors are within 3 years, except for the three highlighted red points for 3 different individuals. For these individuals, we suspect a data error. For one of these individuals their observed age only advances 1 year over 5 waves, for another their age advances 15 years over 2 waves, and for the third their age advances 7 years over 1 wave. **b)** Dropping these three red points, we show that the mean absolute error with this approach is 0.23 years, and the maximum error is 3 years. The histogram shows that for the vast majority of individuals, there is a 0 to 1 year error with this method of deprivatization.
(TIF)

**S16 Fig. DJIN model trained with and without replicated individuals. a)** Time-dependent C-index stratified vs age (points) and for all ages (line). Results are shown for the DJIN model trained without replicated individuals (purple), the DJIN model trained with replicated individuals shown in the main results (red) and a Elastic net Cox model (green). (Higher scores are better). **b)** Brier scores for the survival function vs death age. Integrated Brier scores (IBS) over the full range of death ages is also shown. The Breslow estimator is used for the baseline hazard in the Cox model (Cox-Br). (Lower scores are better). **c)** RMSE scores when the baseline value is observed for each health variable for predictions at least 5 years from baseline, scaled by the RMSE score from the age and sex dependent sample mean (relative RMSE scores). We show the predictions from the DJIN model trained without replicated individuals starting from the baseline value (purple stars), the DJIN model trained with replicated individuals (red circles), predictions assuming a static baseline value (blue diamonds), an elastic-net linear model (green squares). (Lower is better). **d)** Relative RMSE scores when the when the baseline value for each health variable is imputed for predictions past 5 years from baseline. We show the predictions from the DJIN model trained without replicated individuals starting from the imputed value (purple stars), our DJIN model trained with replicated individuals (red circles), and predictions with an elastic-net linear model (green squares).
(TIF)

## Author Contributions

**Conceptualization:** Spencer Farrell.

**Data curation:** Spencer Farrell.

**Formal analysis:** Spencer Farrell.

**Funding acquisition:** Andrew D. Rutenberg.

**Investigation:** Spencer Farrell.

**Methodology:** Spencer Farrell, Andrew D. Rutenberg.

**Project administration:** Andrew D. Rutenberg.

**Resources:** Andrew D. Rutenberg.

**Software:** Spencer Farrell.

**Supervision:** Arnold Mitnitski, Kenneth Rockwood, Andrew D. Rutenberg.

**Visualization:** Spencer Farrell.

**Writing – original draft:** Spencer Farrell.

**Writing – review & editing:** Spencer Farrell, Arnold Mitnitski, Kenneth Rockwood, Andrew D. Rutenberg.

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
