## [Decision Letter · Decision Letter 0]

5 Sep 2021

Dear Dr. Rutenberg,

Thank you very much for submitting your manuscript "Interpretable machine learning for high-dimensional trajectories of aging health" for consideration at PLOS Computational Biology.

As with all papers reviewed by the journal, your manuscript was reviewed by members of the editorial board and by several independent reviewers. In light of the reviews (below this email), we would like to invite the resubmission of a significantly-revised version that takes into account the reviewers' comments.

We cannot make any decision about publication until we have seen the revised manuscript and your response to the reviewers' comments. Your revised manuscript is also likely to be sent to reviewers for further evaluation.

Sincerely,

Tatiana Engel

Guest Editor

PLOS Computational Biology

Mark Alber

Deputy Editor

PLOS Computational Biology

Reviewer's Responses to Questions

**Comments to the Authors:**

Reviewer #1: I congratulate the authors on conducting such interesting and important research and appreciate the efforts they put in developing such a comprehensive model, which I think has a potential to lay a groundwork for expanded applications of ML approaches to research on aging. To me, one of the most appealing features of this model is that the authors postulate that the dynamics of the health trajectories is driven by stochastic differential equations. I believe that incorporating domain knowledge is really a key component in building a successful model so this aspect (using SDE for representing health trajectories) sets this model apart from many existing approaches. Another attractive feature is its multidimensionality, which is well aligned to the studied process of aging and it provides an additional value as illustrated by comparisons with a simplified one-dimensional summary model. I have a few minor comments for authors, mostly about the presentation of the approach and consistency of notations, which, I think, can improve the readability.

1. One part of description of the approach that is not entirely clear to me is section IV.B. In particular, if individuals were “replicated to use all possible starting points” then it seems (the combined likelihood for all individuals was not presented so it is just my (mis-?)interpretation) that the assumption that “replicated individuals are independent in the likelihood” is rather a strong one. If I get it right, this might also contribute to the observed underestimation of standard deviations. Could you please elaborate more on this? I wonder if it would be feasible to check if the approach without such replication of individuals leads to smaller underestimation of st. dev.

2. I appreciate the detailed description of the approach and nice illustrations that provide a lot of useful information about the approach and its performance. I have a few suggestions on the presentation, which I think can further improve the readability: A) As the entire description is dispersed between different places (sections II, IV and Supplement), I would find it useful to repeat some basic details in the beginning of section IV (such as what are indices i, j, k, what are K, L, M, N, etc.) where all the equations are introduced. One example where I was lost is eq. (22)-(23) where it was not clear what is i, t, c^(m1). B) Similarly, I would introduce notations at the first place where they are used (for example, the elementwise multiplication symbol is introduced on p. 9 but first used in eq. (3); notation for the approximate posteriors (q) is explained in Suppl. but not in the main text, etc.). C) Some notations are not consistent throughout the text. For example, theta_P is in eq. (3) in mu_x, but not in the text in the 3rd line on p. 8; h_t is not in eq. (6) but it is later introduced in eq. (21) in IV.E.

3. In Results “population means” are mentioned on a few occasions. As those come from the training set, as I understand, I’d argue that it’s not a population mean, strictly speaking, as this is coming from a subsample of the entire study. Sample means is a better term here in my opinion.

4. Some variables used in the model are discrete ordinal variables with just a few categories. Arguably, the predictions/approximations of those are worse than the continuous ones. Could you elaborate on some recommendations what would be the minimal number of distinct categories for ordinal variables (and the proportion of such variables among all variables) for which the algorithm still gives reasonable results? Say, if one has 10 binary variables, 10 ordinal and 1 continuous, would you still recommend using your method in this case? I also wonder if you check if using only continuous variables from your list improves the performance of the method.

5. Section IV.A mentions that “Ages and death ages for individuals above age 90 are also imputed this way in addition to the year that the wave occurred.” Can you elaborate more on imputation of ages at death? Imputing ages at death is usually not a trivial exercise. Death can occur at any time between the waves. 1-2 years can make a big difference for the ages above 90.

6. In the same section, it is said “Individuals with no recorded death age are considered censored at their last observed age.” It is not clear as the text above suggests that ages at death were imputed. Please clarify.

7. As this algorithm includes both generating trajectories of health states and mortality, I assume the trajectories need to be stopped at the time when the event (mortality) happens for a particular individual. It is not clear from the description whether/how this was implemented in the algorithm.

8. There is inconsistent notation in the text and Suppl. (e.g., Suppl. Table 1 vs. Table I; Suppl. Fig. 1 vs. Fig. S1; “Supplemental Table 2” in IV.F should be Suppl. Table III; in title of Suppl. Table III – “Sec. III of the methods” but there is no Sec. III in the Methods section of the paper).

9. Suppl. Table III is in the middle of Suppl. Figures section. I think it could be moved up to the other suppl. tables, for consistency. In addition, I would find it more convenient if the supplementary figures are reordered to match how they are cited in the text. Otherwise, the readers need to scroll up and down.

10. The text would benefit from an additional proofreading, as there are a few typos. Here is what I caught: “heath” in Abstract and in the Last line on p. 8, “when the when the” in Fig. 2 and Fig. S12; “with with” on p. 4; “that that” on p. 5; “allows us to use add self-supervision” in IV.B.

Reviewer #2: Manuscript describes the development of a probabilistic multidimensional model of aging based on normalized VAE approach. Since the baseline dataset is incomplete, a stochastic linear approach is developed to impute missing data. Predicted aging trajectories reasonably reproduce the baseline and the model performs better than individual linear regressions on each of the individual health variables. Application of VAE to multidimensional longitudinal datasets to model aging trajectory is interesting to explore interactions between health variables and their contribution to aging dynamics. However, there are major concerns of the validity of the approach and the far-reaching conclusion on interpretability that the authors are claiming.

First, the authors have chosen 29 health variables from the ELSA study claiming that their choice was dictated “by availability, but not predictive quality”. However, this choice looks very arbitrary with many if not all chosen variables strongly dependent on each other as, for example, leg raise and walking ability. Many variables are in fact a subset of other variables as, for example, walking ability, ADL and IADL scores, or self-rated health and hearing. Hence the chosen variables do not form an independent 29-multidimensional basis, but perhaps can be described in a drastically reduced and much simpler space.

Second, pair-wise interactions between different variables are assumed to be linear. This is solely done for the purpose of claiming interpretability, since it is straightforward to apply interpretation to a linear weighting. However, since many variables are interdependent (see above) the interpretation is mostly trivial as, for example, connection between levels of glucose and glycated hemoglobin.

Third, some of the chosen variables are sparsely represented in the observed population and are imputed using the stochastic approach. While linear pair-wise relationships derived by the model are interpretable, it is not clear how level of confidence can be assigned to this interpretation. For example, one of the strongest weights in Fig. 3 is that Vitamin D has a negative connection weight with self-rated health. While perhaps an interesting finding, however Vit D measurements are the sparsest of all (as seen from Fig.S1). Imputed VitD distribution within synthetic population is also very far from being close to the observed (Fig.S6).

Fourth, while generating synthetic dataset using the same "imputation" approach that is used to augment the sparse baseline data is perhaps justifiable as a method for sanity checking, however, because of the interdependence of the variables, the observation that the sampled data does "look like" the actual data and, hence, it is properly representing how different variables impact aging is not well grounded.

Fifth, the synthetic dataset is claimed to be useful for training further models when real data are not available. However, this dataset will carry only linear dependencies between interdependent variables that the model learned and, therefore, will be deprived of all other perhaps more interesting interactions.

**Have the authors made all data and (if applicable) computational code underlying the findings in their manuscript fully available?**

Reviewer #1: Yes

Reviewer #2: Yes

PLOS authors have the option to publish the peer review history of their article (what does this mean?). If published, this will include your full peer review and any attached files.

Reviewer #1: No

Reviewer #2: No
---

## [Decision Letter · Decision Letter 1]

11 Dec 2021

Dear Rutenberg,

We are pleased to inform you that your manuscript 'Interpretable machine learning for high-dimensional trajectories of aging health' has been provisionally accepted for publication in PLOS Computational Biology.

Best regards,

Tatiana Engel

Guest Editor

PLOS Computational Biology

Mark Alber

Deputy Editor

PLOS Computational Biology

Reviewer's Responses to Questions

**Comments to the Authors:**

Reviewer #1: Thank you for your detailed response. I have no further comments.

Reviewer #2: The manuscript has improved significantly. Can be accepted.

**Have the authors made all data and (if applicable) computational code underlying the findings in their manuscript fully available?**

Reviewer #1: Yes

Reviewer #2: Yes

PLOS authors have the option to publish the peer review history of their article (what does this mean?). If published, this will include your full peer review and any attached files.

Reviewer #1: No

Reviewer #2: No

---

## [Editor Report · Acceptance letter]

5 Jan 2022

PCOMPBIOL-D-21-01450R1 

Interpretable machine learning for high-dimensional trajectories of aging health

Dear Dr Rutenberg,

I am pleased to inform you that your manuscript has been formally accepted for publication in PLOS Computational Biology. Your manuscript is now with our production department and you will be notified of the publication date in due course.

With kind regards,

Zsofia Freund
